# *FLDmamba:* INTEGRATING FOURIER AND LAPLACE TRANSFORM DECOMPOSITION WITH MAMBA FOR ENHANCED TIME SERIES PREDICTION

## ABSTRACT

Time series prediction, a crucial task across various domains, faces significant challenges due to the inherent complexities of time series data, including non-stationarity, multi-scale periodicity, and transient dynamics, particularly when tackling long-term predictions. While Transformer-based architectures have shown promise, their quadratic complexity with sequence length hinders their efficiency for long-term predictions. Recent advancements in State-Space Models, such as Mamba, offer a more efficient alternative for long-term modeling, but they lack the capability to capture multi-scale periodicity and transient dynamics effectively. Meanwhile, they are susceptible to the data noise issue in time series. This paper proposes a novel framework, **FLDmamba** (**F**ourier and **L**aplace Transform **D**ecomposition **Mamba**), addressing these limitations. **FLDmamba** leverages the strengths of both Fourier and Laplace transforms to effectively capture both multi-scale periodicity, transient dynamics within time series data, and improve the robustness of the model to the data noise issue. Our extensive experiments demonstrate that **FLDmamba** achieves superior performance on time series prediction benchmarks, outperforming both Transformer-based and other Mamba-based architectures. This work offers a computationally efficient and effective solution for long-term time series prediction, paving the way for its application in real-world scenarios. To promote the reproducibility of our method, we have made both the code and data accessible via the following URL: https://anonymous.4open.science/r/FLDmamba.

## 1 INTRODUCTION

Time series prediction, which forecasts the future values of a (multivariate) variable based on its historical values, finds its application across a wide range of fields. Examples include weather prediction (Lorenc, 1986; Bauer et al., 2015), power grid management (Tang, 2011), traffic prediction (Yu et al., 2017; Bai et al., 2020), and stock market (Fama, 1970), to name just a few. Despite significant advancements in this domain, the inherent complexities of time series data, such as non-stationarity, multi-scale periodicity, intrinsic stochasticity, and noise, pose substantial challenges to existing predictive models in long-term prediction.

Transformer-based architectures (Vaswani et al., 2017), successful in NLP and computer vision, have been explored extensively in time series prediction. Although they demonstrate impressive performance, they face degraded accuracy and efficiency in long-term time series prediction due to their quadratic complexity w.r.t. sequence length. iTransformer (Liu et al., 2023) addresses inter-series dependencies by inverting attention layers, but its tokenization approach, which uses a simple MLP layer, fails to capture intricate evolutionary patterns in the data, as shown in Figure 1. Thus, Transformer-based models face challenges in computational efficiency and predictive performance. This can be explained by that the computational cost of self-attention mechanism, which is at the heart of Transformer-based model, is $\mathcal{O}(L^2)$, where $L$ is the sequence length. Meanwhile, the self-attention mechanism leads to point-wise treatment independently and failure to capture intricate evolutionary patterns in time series data.

Recently, architectures based on State-Space Models (Gu et al., 2021a; Smith et al., 2022) have emerged as a promising alternative due to the computational efficiency inherent in linear models to address the long-term prediction challenge. A notable example is Mamba (Gu & Dao, 2023), which employs a linear state space with input-dependent selection. The linear state space allows efficient and parallelized long-sequence modeling, while the input-dependent selection allows propagating and forgetting information in long sequences, facilitating in-context learning. Mamba's design is a good start. However, there are three challenges that Mamba-based methods for time-series prediction cannot address. **(1) Multi-scale periodicity**. Time series data typically consists of patterns that occur periodically, such as in traffic, electricity, and weather. In addition, the periodic patterns typically exist in multiple time scales and are superimposed together. For example, in weather data, the temperature can fluctuate both in the time scale of a day and a year. Mamba lacks frequency modeling to capture such multi-scale periodicity. **(2) Transient dynamics**. In addition to periodicity, time series data often shows complex transient dynamics, which can be characterized as time-varying patterns, short-term fluctuations, or event-driven variations. These transient dynamics pose significant challenges for Mamba, as Mamba exhibits a tendency to prioritize point-wise temporal dynamics over neighboring transient dynamics. Figure 1 presents a comparative analysis of the time series predicted by S-Mamba against the ground truth values on the real-world datasets ETTm1 and ETTm2 (Zhou et al., 2021). A visual inspection reveals a distinct disparity in the distribution of the predicted time series compared to the ground truth. This discrepancy is due to that S-Mamba fails to effectively capture multi-scale periodicity and transient dynamics inherent within the time series data. **(3) Data noise.** Noise in time series data introduces random fluctuations into the data, increasing the uncertainty in predictions. Models trained on noisy data may produce less reliable forecasts with wider prediction intervals, making it harder to make accurate predictions.

To address the limitations of existing methods, this paper proposes two key technical advancements to enhance Mamba in time series prediction: **(1) Incorporating Fourier analysis into Mamba:** Mamba primarily focuses on capturing temporal dynamics in the temporal domain, lacking the ability to model long-term dynamics in the frequency domain, such as multi-scale patterns overlooked in the temporal domain. To address this, we propose integrating Fourier analysis into Mamba, enabling it to capture long-term properties, such as multi-scale patterns, in the frequency domain. In addition, the Fourier Transform can help in separating the underlying patterns or trends from noise in the time series data by highlighting dominant frequency components. By focusing on these dominant frequencies, the model can reduce the impact of noise that might otherwise affect the accuracy of predictions, thereby enhancing the model's robustness to noisy data. **(2) Integrating Laplace analysis into Mamba:** To improve Mamba's ability to capture transient dynamics, such as short-term fluctuations, we introduce Laplace analysis into Mamba. This integration allows the model to better understand the relationships between neighboring data points and capture transient changes. Based on these two advancements, we propose a novel framework, **FLDmamba** (**F**ourier and **L**aplace Transform **D**ecomposition **Mamba**), specifically designed for long-term time series prediction. **FLDmamba** leverages the strengths of both Fourier and Laplace analysis, enabling it to effectively capture both multi-scale periodicity and transient dynamics within time series data.

The core innovation of FLDmamba lies in its strategic integration of frequency analysis and Laplace analysis within the Mamba framework. By representing time series data in the frequency domain through Fourier analysis, FLDmamba effectively captures multi-scale periodicity, improving the ability to conduct long-term prediction. Simultaneously, the incorporation of Laplace analysis improves the model's capacity to capture local correlations between neighboring data points, leading to a more accurate representation of transient dynamics. As shown in Figure 1, FLDmamba significantly outperforms S-Mamba. In addition, as the backbone of our framework FLDmamba is Mamba, it is highly efficient and well-suited for deployment in large-scale real-world applications.

We summarize our contributions as follows:

- **An Efficient Unified Framework for Long-term Time Series Prediction**. We present an efficient and unified framework for long-term time series prediction that eliminates the need for feature engineering.

- **Enhanced by Fourier and Laplace Transformations, Decomposed Mamba excels in capturing multi-scale periodicity, transient dynamics, and mitigating noise.** Through the integration of the Fourier and Laplace Transforms into Mamba, our proposed model, FLDmamba, adeptly captures intricate multi-scale periodic patterns and dynamic fluctua-

tions present in time series data. This approach not only diminishes the impact of noise but also fortifies the model's resilience, culminating in a substantial enhancement in long-term time series prediction accuracy.

- **Extensive Experiments.** Evaluated on time series prediction benchmarks and comparing with strong baselines including transformer-based and other Mamba-based architectures, our FLDmamba achieves state-of-the-art (SOTA) performance on of tasks.

## 2 RELATED WORK

**Time Series Prediction**. Time series prediction, forecasting future values based on historical data (Lim & Zohren, 2021; Torres et al., 2021), has witnessed a surge in advancements driven by deep neural network techniques. Notably, Mamba (Gu & Dao, 2023) and Transformer (Vaswani et al., 2017) have emerged as prominent players in this domain, achieving notable successes in time series prediction (Patro & Agneeswaran, 2024; Liang et al., 2024; Vaswani et al., 2017). Transformer-based methods, in particular, have garnered significant attention due to their self-attention mechanism (Vaswani et al., 2017), which enables the capture of long-range dependencies within time series data. However, the quadratic complexity inherent in the Transformer architecture presents a formidable challenge for long-term time series

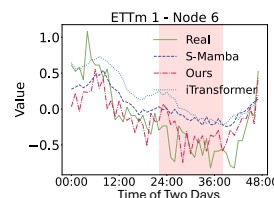

Figure 1: Time Series Distributions of ground truth, S-Mamba, iTransformer and Ours.

**R#8Y8C-W6**

prediction. The computational burden associated with processing lengthy sequences significantly hinders the model's performance, particularly when dealing with extended time horizons. This challenge has spurred researchers to explore innovative approaches that balance computational efficiency with predictive accuracy. One such approach, proposed by (Liu et al., 2021), introduces a pyramidal attention module that effectively summarizes features at different resolutions. FEDformer (Zhou et al., 2022) leverages a frequency domain enhanced Transformer architecture to enhance both efficiency and effectiveness. Zhang & Yan (2022) further contribute to this field with Crossformer, which incorporates a patching operation, similar to other models, but also employs Cross-Dimension attention to capture dependencies between different time series. While patching reduces the number of elements to be processed and extracts comprehensive semantic information, these models still face limitations in performance when handling exceptionally long sequences. A recent work proposes Moirai (Woo et al., 2024), which pretrains a model with large-scale datasets. It has different settings from other existing full-shot studies. Thus, It is out-of-scope for our baselines.

**R#RZwJ-Q1**

To address this persistent challenge, iTransformer (Liu et al., 2023) introduces an innovative approach that inverts the attention layers, enabling the model to effectively capture inter-series dependencies. However, iTransformer's tokenization strategy, which simply passes the entire sequence through a Multilayer Perceptron (MLP) layer, fails to adequately capture the intricate evolutionary patterns inherent in time series data. This limitation underscores the ongoing need for more sophisticated techniques that can effectively model the complex dynamics of time series data. More related work on mamba-based methods for time series prediction is shown in Appendix 6.3.

**R#vEmK-W6**

In conclusion, while Transformer-based models have demonstrated significant promise in time series prediction, they still grapple with challenges related to computational efficiency and performance when dealing with long sequences. Continued research efforts are crucial to developing more efficient and effective architectures that can effectively model the intricate complexities of time series data, ultimately paving the way for more accurate and reliable long-term predictions.

**Models based on SSMs (State-Space Models)**. Previous approaches to time series prediction, such as those found in AGCRN (Bai et al., 2020), DCRNN (Li et al., 2018), and ASTGCN (Guo et al., 2019), primarily relied on recurrent neural networks (RNNs) (Sutskever et al., 2014) or convolutional neural networks (CNNs) (Krizhevsky et al., 2017). RNN-based methods process sequential data in a step-by-step manner, propagating gradients cell-by-cell, which can hinder training speed and limit the retention of long-term information. Conversely, CNN-based methods employ convolutional kernels to capture local information, resulting in reduced inference speed and overlooking long-term global information. To address these limitations, a novel state-space model called Mamba was introduced in Gu & Dao (2023). Mamba aims to capture long-term information while maintaining computational efficiency. Building upon the foundation laid by Mamba (Gu & Dao,

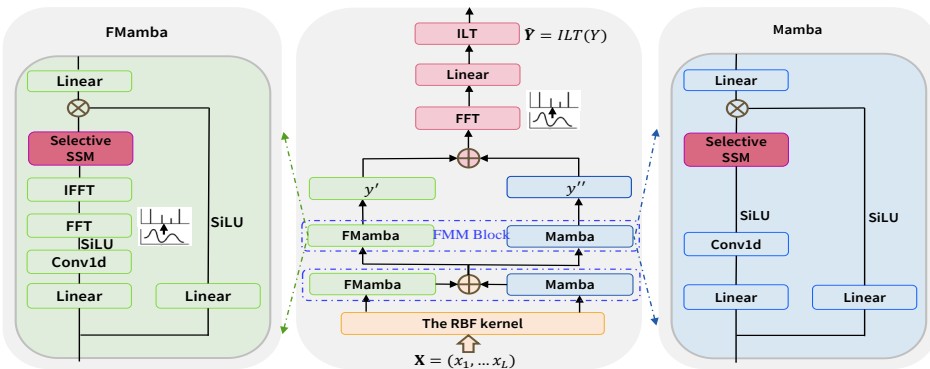

Figure 2: This diagram illustrates the architecture of FLDmamba, showcasing the individual components and their integration. **Left**: This section provides a detailed view of the FMamba architecture, highlighting its key components and their interactions. **Middle:** The central section presents the overall architecture of FLDmamba, demonstrating how FMamba, the Fourier and Laplace Transform modules, and Mamba are interconnected to form the complete framework. **Right:** The rightmost section focuses on the architecture of Mamba, providing a visual representation of its internal structure and operation.

2023), we propose FLDmamba, which leverages the power of Fourier and Laplace transforms. The incorporation of the Fourier transform in Mamba's input-selection stage facilitates the capture of multi-scale periodicity, while the Laplace transform-powered output module explicitly models periodic and transient dynamics. This strategic integration enhances the model's ability to capture both long-term dependencies and complex temporal patterns. The integration of Fourier and Laplace transforms into the Mamba framework in FLDmamba represents a significant advancement in time series prediction. By leveraging these powerful mathematical tools, our model surpasses the limitations of previous RNN and Trasformer-based approaches, enabling more accurate and efficient time series forecasting performance.

## 3 METHODOLOGY

This section details the FLDmamba framework (illustrated in Figure 14), which comprises five components: (1) data smoothing using a radial basis function kernel; (2) an FMamba encoder layer (using Fast Fourier Transform for multi-scale periodic pattern extraction); (3) a Mamba encoder layer for modeling long-term dependencies; (4) an integrated FMamba-Mamba block capturing both periodic and transient dynamics and separating data noise; (5) and an inverse Laplace transform to produce time-domain predictions. The preliminary background (Mamba, Fourier/Laplace Transforms) is shown in Appendix 6.2. Then FLDmamba's computational complexity is analyzed (shown in Appendix 6.4). Subsequent sections offer a detailed breakdown of each component. Firstly, the problem definition is shown as follows:

**Problem Statement**. Given the input with the long-term time series data $\mathbf{X} = (x_1, ...x_L) \in \mathbb{R}^{L \times V}$, where $L$ is the size of history window and $V$ is number of variates, the ground truth of the predicted output is $\mathbf{Y}^{(1)} = (x_{L+1}, ...x_{L+H}) \in \mathbb{R}^{H \times V}$, where $H$ is the prediction size of future time steps. We aim to learn a mapping function $\mathcal{F}$ to satisfy $\widehat{\mathbf{Y}} = \mathcal{F}(\mathbf{X})$ and minimize the loss $\frac{1}{|\mathbf{Y}^{(1)}|} \sum_{i=1}^{|\mathbf{Y}^{(1)}|} (\hat{y}_i - y_i^{(1)})^2$, where temporal dependencies are preserved.

### 3.1 FLDMAMBA

Our proposed approach, FLDmamba, is illustrated as follows: the Radial Basis Function (RBF) kernel, the FMamba encoder layer enhanced by the Fast Fourier transform (FFT), the Mamba encoder layer, the FMM block, and the inverse Laplace transform (ILT) module for FLDmamba. Each serves a specific purpose in the overall framework. In the following sections, we provide comprehensive explanations and illustrations for each of these components, outlining their respective functionalities and contributions within the FLDmamba framework. For a comprehensive understanding of the algorithm's steps, please refer to Algorithm 1 in Appendix 6.1.

### 3.1.1 DATA SMOOTHING VIA THE RADIAL BASIS FUNCTION KERNEL

To achieve data smoothing on the input data matrix $\mathbf{X}$, we propose the utilization of the radial basis function (RBF) kernel. The RBF kernel is a widely employed mathematical function in machine learning algorithms, specifically for tasks such as prediction. Its primary purpose is to facilitate the effective capture of intricate temporal relationships and patterns within time series data. The RBF kernel for the data point $x_o$ is mathematically defined as follows:

$$x'_o := \mathcal{K}(x_o, x_p) = \exp(-\gamma||x_o - x_p||^2) = \langle \varphi(x_o), \varphi(x_p) \rangle$$
$$\approx \langle z(x_o), z(x_p) \rangle; \ o \neq p \tag{1}$$

In this equation, $x'_o$ is the output of the RBF kernel on $x_o$. $\mathcal{K}$ denotes the kernel function, $x_o$ and $x_p$ represent input data points, $\gamma = \frac{1}{2\sigma'^2}$ is a hyperparameter that controls the width of the kernel, and $|| \cdot ||$ denotes the Euclidean distance between the points, and we suppose $\sigma' = 1$ in this paper. Additionally, $\varphi(x_o)$ is defined as $\exp(-\frac{1}{2}||x_o||^2)(a_{q-0}^{(0)}, ..., a_1^{(j)}, ..., a_{q_j}^{(j)}, ...)$, where $q_j = \binom{k+j-1}{j}$ and $a_q^{(j)} = \frac{x_1^{n_1}...x_k^{n_k}}{\sqrt{n_1!...n_k!}}$, with $n_1 + n_2 + ... + n_k = j$ and $1 \leq q \leq q_j$. The symbol $\wedge$ represents the exterior product. Moreover, the function $z$ maps a single vector to a high-dimensional vector that approximates the RBF kernel. To construct this function $z$, we randomly sample from the Fourier transform of the kernel, denoted as $\phi(x_o) = \frac{1}{\sqrt{r}}[\cos\langle w_1, x_o \rangle, \sin\langle w_1, x_o \rangle, ..., \cos\langle w_r, x_o \rangle, \sin\langle w_r, x_o \rangle]^T$, where $w_1, ..., w_r$ are independent samples drawn from the Gaussian distribution $\mathcal{N}(0, \sigma'^{-2}I)$.

### 3.1.2 FMAMBA ENCODER LAYER POWERED BY THE FAST FOURIER TRANSFORM (FFT)

To capture multi-scale periodicity, *e.g.*, daily and monthly patterns, and alleviation data noise, we propose to adopt the Fourier transform to endow state space models on the step size $\Delta \in \mathbb{R}^{2L \times V}$ to filter different periodic patterns out from noise, which is hard to address by existing time-series methods like S-Mamba (Wang et al., 2024) and iTransformer (Liu et al., 2023). In this section, we aim to illustrate the Fourier transform-powered FMamba encoder. As we know from the preliminary in Section 6.2 in Appendix, an important input-dependent selection mechanism is how the step size $\Delta$ is dependent on the input. However, all information of the input is passed through $\Delta$ at each time step without filtering, which has three drawbacks. Firstly, not all information obtained by this selective mechanisms is important. Secondly, after projection via this selective mechanism, the periodic patterns in time series data are hard to capture. Thirdly, noise in data is hard to be distinguished by $\Delta$. Motivated by the above reasons, we propose to adopt the Fourier transform on the $\Delta$ to identify important frequency information and further capture the multi-scale periodic patterns in time series data. Firstly, we define a kernel integral operator, which aims to identify relevant information by convolving the input signal $x$ from the previous layer with a kernel $\tilde{\mathcal{K}}(\Delta t; \phi)$ with time difference $\Delta t$ and parameter $\phi$:

**Definition 1: (Kernel integral operator)** We define the kernel integral operator $\mathcal{I}(x; \phi)$ as follows:

$$\mathcal{I}(x; \phi)(t) = \int_D \tilde{\mathcal{K}}(t - s; \phi) x_s ds \tag{2}$$

R#Xe6T-O1

Here $t, s$ denote time. The convolution theorem states that the Fourier transform $\mathcal{F}$ applied to the above kernel integral operator, can be expressed as the product of the Fourier transform of the kernel and the Fourier transform of the input signal. Therefore,

$$\mathcal{I}(x; \phi)(t) = \mathcal{F}^{-1}(\tilde{W} \cdot \mathcal{F}(x)) \tag{3}$$

Here $\mathcal{F}^{-1}$ is the inverse Fourier transform, $\tilde{W}$ is the Fourier transform of the kernel $\tilde{\mathcal{K}}$, and we directly treat $\tilde{W}$ as a learnable parameter matrix. The functionality of the kernel $\tilde{\mathcal{K}}$ is to identify relevant signals and filter out noise. In addition, to improve the efficiency of operation, Fast Fourier Transform (FFT) is adopted for the above $\mathcal{F}$. For the Fourier transform of the input signal $x$, we define $D = \mathcal{F}(x) \in \mathbb{R}^{2L \times V}$ for each feature $j$ of $x$ as:

$$D_j[k] = \mathcal{F}_j(k) = \sum_{n=1}^{L} x_{nj} \cdot e^{-\hat{i}\frac{2\pi}{L}kn}; j \in [1, 2, ..V]; \hat{i} = \sqrt{-1} \tag{4}$$

$D_j[k] \in \mathbb{C}^{d_f}$ is the Fourier transform of the $j$-th variable at frequency index $k$ and $d_f$ represents the sequence length after FFT in frequency domain. And $\hat{i}$ denotes the imaginary unit. Then we

transform it into temporal domain via Inverse FFT (IFFT), producing $\Delta_F$, which is the filtered version of $\Delta$, via the kernel integral operator $\mathcal{I}(x; \phi)$ defined above:

$$\Delta_F(n, j) := \mathcal{I}(x_j; \phi)(n) = \frac{1}{L} \sum_{k=1}^{L} \tilde{W} \cdot D_j[k] \cdot e^{\hat{i}\frac{2\pi}{L}kn}; j \in [1, 2, ..V]; \hat{i} = \sqrt{-1} \tag{5}$$

$$\bar{\mathbf{A}}_F = \exp(\Delta_F \mathbf{A}); \ \bar{\mathbf{B}}_F = \Delta_F \mathbf{A}^{-1} \exp(\Delta_F \mathbf{A}) \cdot \Delta_F \mathbf{B}$$

The filtered $\Delta_F$ replaces the $\Delta$ in the original Mamba, and can better capture relevant and periodic information in the presence of noise. Based on the output $\mathbf{X}'$ of the RBF kernel, we can obtain the final output as follows:

$$u_i^{(1)} \leftarrow \text{SSM}(\bar{\mathbf{A}}_F, \bar{\mathbf{B}}_F, \mathbf{C})(x_i'); \ u_i^{(2)} \leftarrow u_i^{(1)} \otimes \text{SiLU}(\text{Linear}(x_i')); \ u_i \leftarrow \text{Linear}(u_i^{(2)}) \tag{6}$$

Where $x_i' \in \mathbb{R}^V$ denotes the output via the RBF at the time step $i$. SiLU denotes the activation function. And Linear represents the linear layer. And $u_i^{(1)} \in \mathbb{R}^V$, $u_i^{(2)} \in \mathbb{R}^V$ and $u_i \in \mathbb{R}^V$ are three outputs. A detailed algorithm is shown in Algorithm 2 in Appendix 6.1.

### 3.1.3 MAMBA ENCODER LAYER

To capture long-term dependencies in time-series sequences, we incorporate Mamba into our framework, working in parallel with FMamba. Unlike the multi-head attention mechanism in Transformer, Mamba employs a selective mechanism to model feature interactions. The core concept of Mamba is to map the input sequence $\mathbf{X}' = (x_1', x_2', \ldots, x_L')$ to the output $U'$ through a hidden state $h(i)$, which acts as a linear time-invariant system. More specifically, given the input sequence $x_i' \in \mathbb{R}^V$, where $V$ represents the number of variables in the time series data, we utilize Mamba to model it (Gu et al., 2021b). The process of Mamba can be outlined as $h'(i) = \mathbf{A}h(i) + \mathbf{B}x_i', i \in [1, L]$. Here, $x_i' \in \mathbb{R}^V$. The discretized process, represented by $\Delta$, can be illustrated as follows:

$$\bar{\mathbf{A}} = \exp(\Delta \mathbf{A}); \ \bar{\mathbf{B}} = \Delta \mathbf{A}^{-1} \exp(\Delta \mathbf{A}) \cdot \Delta \mathbf{B} \tag{7}$$

Then, we can obtain the output via the Mamba encoder layer $U' \in \mathbb{R}^{L \times V}$ as follows:

$$u_i'^{(1)} \leftarrow \text{SSM}(\bar{\mathbf{A}}, \bar{\mathbf{B}}, \mathbf{C})(x_i'); \ u_i'^{(2)} \leftarrow u_i'^{(1)} \otimes \text{SiLU}(\text{Linear}(x_i')); \ u_i' \leftarrow \text{Linear}(u_i'^{(2)}) \tag{8}$$

Where $u_i'^{(1)} \in \mathbb{R}^V$, $u_i'^{(2)} \in \mathbb{R}^V$ and $u_i' \in \mathbb{R}^V$ are three outputs at the time step $i$. A detailed algorithm is shown in Algorithm 3 in Appendix 6.1.

### 3.1.4 THE FMAMBA-MAMBA (FMM) BLOCK FOR FLDMAMBA

Based on the concepts of FMamba and Mamba, we propose the integration of these two components into a single block, which we refer to as the FMamba-Mamba (FMM) block. Drawing inspiration from the ResNet mechanism (He et al., 2016), an FMM block consists of a FMamba encoder and a Mamba encoder in parallel, both sharing the same input and whose outputs are summed together, producing the output of the FMM block. In this way, it can effectively capture the intricate temporal and periodic dependencies present in the data. Subsequently, the output of the first FMM block is passed to a second FMM block (whose output of the second FMamba is $y'$ and output of second Mamba is denoted as $y''$ in Figure 14). The process is illustrated as follows:

$$u_i'' = u_i' + u_i; \ y_i' \leftarrow \text{FMamba encoder layer}(u_i''); \ y_i'' \leftarrow \text{Mamba encoder layer}(u_i'');$$
$$Y_i \leftarrow \text{Linear}(\text{FFT}(y_i' + y_i'')); \tag{9}$$

Where $u_i \in \mathbb{R}^V$ and $u_i' \in \mathbb{R}^V$ denote outputs of the time step $i$ of the first-layer FMamba and the first-layer Mamba respectively. $y_i'' \in \mathbb{R}^V$, $y_i' \in \mathbb{R}^V$ and $Y_i \in \mathbb{R}^V$. A detailed description of this process can be found in Figure 14 and Algorithm 1. To assess the impact and effectiveness of the FMM block, we conducted experiments and present the results in the ablation study.

### 3.1.5 INVERSE LAPLACE TRANSFORM FOR FLDMAMBA

There are many transient dynamics factors in time series data that hamper the performance of existing methods. Meanwhile, we also aim to capture long-term periodic patterns that are hard to capture

in time series data by existing methods. Due to the success of Laplace transform on many domains (Camacho et al., 2019), we propose to adopt the inverse Laplace transform (ILT) on them, which is able to capture transient dynamics and long-term periodic patterns. It is shown as: $\hat{Y}(t) = \frac{1}{2\pi\hat{i}} \lim_{T\to\infty} \int_{\gamma-\hat{i}T}^{\gamma-\hat{i}T} K_\phi(s)Y(s)e^{st}ds; \quad \hat{i} = \sqrt{-1}$, where $Y(s)$ is the Laplace transform of $Y(t)$ from the previous layer. And $K_\phi(s)$ is a kernel in the Laplace domain. By stipulating first-order singularities as $K_\phi(s) = \sum_{n=1}^{N} \frac{\beta_n}{s-\mu_n}$, we derive in Appendix 6.2 that **R#6yv6-W1**

$$\hat{Y}(t) = \sum_{n=1}^{M} A_n e^{-\sigma_n t} \cos(w_n t + \varphi_n) \tag{10}$$

where $A_n, \xi_n, w_n$, and $\phi_n$ are all functions of $Y(t)$ and the $\{\beta_n\}$ and $\{\mu_n\}$. Thus in our work, we directly parameterize $A_n, \xi_n, w_n$, and $\phi_n$ as learnable functions of $Y(t)$ from the previous layer to improve efficiency and stability. We see in Eq. 10, the cosine term $\cos(w_n t)$ plays a crucial role in capturing the periodicity inherent in the data. It is capable of effectively identifying and modeling recurring patterns or cycles within the time series. On the other hand, the term $e^{\sigma_n t}$ is responsible for capturing the transient dynamics exhibited by the data. It enables the model to capture and represent the short-lived variations or irregularities in the time series. Besides, the combined use of exponential $e^{\sigma_n t}$ and $\cos(w_n t)$ terms ensures that the reconstructed time-domain data accurately reflects both transient dynamics and long-term periodic trends, making it suitable for forecasting future behaviors based on historical data. This, in turn, contributes to improved accuracy and predictive capabilities, allowing the model to make more reliable forecasts and capture the nuances of the data more effectively. Model complexity is shown in Appendix 6.4.

## 4 EVALUATION

In this section, we aim to conduct experiments to answer the following questions: **Q1:** What is the effectiveness of FLDmamba compared with other state-of-the-art baselines? **Q2:** How each component of FLDmamba affect the final performance? **Q3:** How is the robustness of FLDmamba compared with state-of-the-art methods like S-Mamba and iTransfomrer? **Q4:** How is the advantage of FLDmamba on long-term prediction with increasing lookback length compared to other state-of-the-art methods? **Q5:** How is the performance of FLDmamba on capturing multi-scale periodicity and transient dynamics compared to state-of-the-art baselines? **Q6:** How is the efficiency of FLD-mamba compared to state-of-the-art baselines? (in Appendix 6.5) **Q7:** How do hyperparameters of FLDmamba affect the performance (in Appendix 6.5)?

### 4.1 EXPERIMENTAL SETUP

**Datasets**. To rigorously evaluate the effectiveness of our proposed model, we selected a diverse set of 9 real-world datasets (Zhou et al., 2023; 2021) for evaluation. These datasets encompass a range of domains, including Electricity, 4 ETT datasets (ETTh1, ETTh2, ETTm1, ETTm2), and others. These datasets are extensively utilized in research and span various fields, such as transportation analysis and energy management. Detailed statistics for each dataset can be found in Table 2 in Appendix 6.5.

**Baselines**. We compare our method FLDmamba with 10 state-of-the-art methods including 6 Transformer-based models, 3 MLP-based methods and 1 SSM-based method. The detailed illustrations and experiment settings are shown in Appendix 6.5.

### 4.2 OVERALL COMPARISON (Q1)

We present evaluation results using two metrics: Mean Squared Error (MSE) and Mean Absolute Error (MAE) (Table 1). Based on results, we make the following observations:

**Outstanding Performance**. Our proposed framework, FLDmamba, demonstrates exceptional performance across a range of time series prediction tasks. As shown in Table 1, FLDmamba achieves state-of-the-art results in the majority of scenarios (60 out of 72, or 83.3%), and consistently ranks among the top performers in the remaining cases across nine real-world datasets. This outstanding performance can be attributed to several key design elements: **(1) Data Smoothing via the Radial**

**Basis Function (RBF) Kernel:** FLDmamba incorporates an RBF kernel, which effectively smooths the input data, reducing noise and enabling more accurate capture of underlying temporal patterns. This data preprocessing step significantly contributes to the model's improved prediction accuracy. **(2) Multi-Scale Periodicity Capture with the Fast Fourier Transform (FFT):** Our framework incorporates the FFT on the parameter $\Delta$. This transformation enables the identification and extraction of multi-scale periodic patterns present in the time series data. By effectively capturing these periodic patterns, FLDmamba significantly enhances its predictive capabilities. **(3) Enhanced Long-Term Prediction and Transient Dynamics Capture with the Inverse Laplace Transform:** To further improve long-term predictions and capture transient dynamics, FLDmamba incorporates the inverse Laplace transform on the combined outputs of FMamba and Mamba. This innovative approach proves advantageous in capturing both transient dynamics and periodic patterns, further boosting the accuracy of our prediction outputs. **(4) Integration of FMamba and Mamba via the FMM Block:** The FMM block within FLDmamba facilitates the capture of complex temporal attributes and dependencies between the FMamba and Mamba components. This integration enhances the model's ability to capture intricate temporal relationships, improving overall performance.

The superior performance FLDmamba in time series prediction arises from the synergistic combination of an RBF kernel, Fast Fourier and inverse Laplace transforms, and the integrated FMamba and Mamba components. This approach effectively captures temporal patterns, multi-scale periodicity, transient dynamics, and mitigates noise.

## 4.3 ABLATION STUDY (Q2)

This section aims to evaluate the individual contributions of each component within our proposed framework, FLDmamba, as illustrated in Figure 3 and Figure 8 (Appendix 6.5). We conduct an ablation study by considering five variants: **"w/o FT":** This variant excludes the Fourier transform for the parameter $\Delta$, allowing us to assess the impact of frequency domain analysis. **"w/o FM":** This variant re-

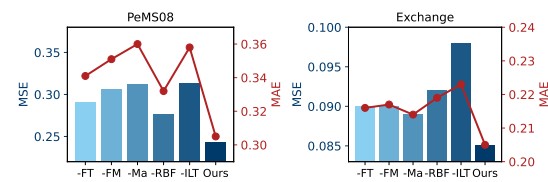

Figure 3: Ablation study of FLDmamba with $L = 96$.

moves the FMamba component, leaving only the Mamba architecture, enabling us to evaluate the contribution of the frequency-domain enhanced Mamba. **"w/o Ma":** This variant eliminates the Mamba component, retaining only FMamba, allowing us to assess the impact of the frequency-domain modeling. **"w/o RBF":** This variant omits the Radial Basis Function (RBF) kernel, enabling us to evaluate the impact of data smoothing on performance. **"w/o ILT":** This variant disregards the inverse Laplace transform, allowing us to assess the impact of the time-domain conversion.

By comparing the performance of these variants against our full method, FLDmamba, we can isolate the individual contribution of each component to overall performance. The results presented in Figure 3 and Figure 8 demonstrate that each component of FLDmamba positively influences performance, confirming the effectiveness of our approach. Notably, the inverse Laplace transform exhibits the most significant impact on the overall effectiveness of our method FLDmamba.

## 4.4 ROBUSTNESS (Q3)

This study investigates the robustness of our proposed method, FLDmamba, in comparison to S-Mamba and iTransformer, under conditions of noisy time-series data. The experiments were conducted on the ETTm1 dataset, where varying levels of noise (specifically, 10% and 15%) were systematically introduced into the test datasets. The results, visualized in Figure 4, reveal a clear performance advantage for FLDmamba across both noise levels. Further-

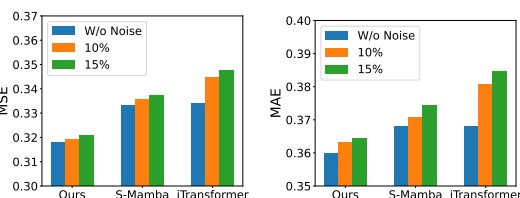

Figure 4: Performance comparison of robustness.

more, a key finding is that the performance decrement observed in FLDmamba is significantly smaller than that of the competing methods as the noise intensity increases. This empirically validates the inherent robustness of our method in mitigating the adverse effects of noise. The superior

R#7Gt7-Q6

Table 1: We present comprehensive results of FLDmamba and baselines on the ETTh1, ETTh2, Electricity, Exchange, Weather, and Solar-Energy datasets. The lookback length $L$ is fixed at 96, and the forecast length $T$ varies across 96, 192, 336, and 720. Bold font denotes the best model and underline denotes the second best.

R#vEmK-W5, R#6yv6-W2

| Models | | FLDmamba (Ours) | | S-Mamba | | SST | | Bi-Mamba+ | | iTransformer | | RLinear | | PatchTST | | Crossformer | | TiDE | | TimesNet | | DLinear | | FEDformer | | Autoformer | |
|---|---|---|---|---|---|---|---|---|---|---|---|---|---|---|---|---|---|---|---|---|---|---|---|---|---|---|---|
| Metric | | MSE | MAE | MSE | MAE | MSE | MAE | MSE | MAE | MSE | MAE | MSE | MAE | MSE | MAE | MSE | MAE | MSE | MAE | MSE | MAE | MSE | MAE | MSE | MAE | MSE | MAE |
| ETTm1 | 96 | **0.318** | **0.360** | 0.333 | 0.368 | 0.337 | 0.374 | 0.355 | 0.386 | 0.334 | 0.368 | 0.355 | 0.376 | 0.329 | 0.367 | 0.404 | 0.426 | 0.364 | 0.387 | 0.338 | 0.375 | 0.345 | 0.372 | 0.379 | 0.419 | 0.505 | 0.475 |
| | 192 | **0.365** | **0.384** | 0.376 | 0.390 | 0.377 | 0.392 | 0.415 | 0.419 | 0.377 | 0.391 | 0.391 | 0.392 | 0.367 | 0.385 | 0.450 | 0.451 | 0.398 | 0.404 | 0.374 | 0.387 | 0.380 | 0.389 | 0.426 | 0.441 | 0.553 | 0.496 |
| | 336 | **0.404** | **0.409** | 0.408 | 0.413 | 0.401 | 0.412 | 0.450 | 0.442 | 0.426 | 0.420 | 0.424 | 0.415 | 0.399 | 0.410 | 0.532 | 0.515 | 0.428 | 0.425 | 0.410 | 0.411 | 0.413 | 0.413 | 0.445 | 0.459 | 0.621 | 0.537 |
| | 720 | **0.464** | 0.441 | 0.475 | 0.448 | 0.498 | 0.464 | 0.497 | 0.476 | 0.491 | 0.459 | 0.487 | 0.450 | **0.454** | **0.439** | 0.666 | 0.589 | 0.487 | 0.461 | 0.478 | 0.450 | 0.474 | 0.453 | 0.543 | 0.490 | 0.671 | 0.561 |
| | Avg | 0.389 | 0.399 | 0.398 | 0.405 | 0.413 | 0.411 | 0.429 | 0.431 | 0.407 | 0.410 | 0.414 | 0.407 | **0.387** | **0.400** | 0.513 | 0.496 | 0.419 | 0.419 | 0.400 | 0.406 | 0.403 | 0.407 | 0.448 | 0.452 | 0.588 | 0.517 |
| ETTm2 | 96 | **0.173** | **0.253** | 0.179 | 0.263 | 0.185 | 0.274 | 0.186 | 0.278 | 0.180 | 0.264 | 0.182 | 0.265 | 0.175 | 0.259 | 0.287 | 0.366 | 0.207 | 0.305 | 0.187 | 0.267 | 0.193 | 0.292 | 0.203 | 0.287 | 0.255 | 0.339 |
| | 192 | **0.240** | **0.299** | 0.250 | 0.309 | 0.248 | 0.313 | 0.257 | 0.324 | 0.250 | 0.309 | 0.246 | 0.304 | 0.241 | 0.302 | 0.414 | 0.492 | 0.290 | 0.364 | 0.249 | 0.309 | 0.284 | 0.362 | 0.269 | 0.328 | 0.281 | 0.340 |
| | 336 | **0.301** | **0.307** | 0.312 | 0.349 | 0.309 | 0.351 | 0.318 | 0.362 | 0.311 | 0.348 | 0.307 | 0.342 | 0.305 | 0.343 | 0.597 | 0.542 | 0.377 | 0.422 | 0.321 | 0.351 | 0.369 | 0.427 | 0.325 | 0.366 | 0.339 | 0.372 |
| | 720 | **0.401** | **0.397** | 0.411 | 0.406 | 0.406 | 0.405 | 0.412 | 0.416 | 0.412 | 0.407 | 0.407 | 0.398 | 0.402 | 0.400 | 1.730 | 1.042 | 0.558 | 0.524 | 0.408 | 0.403 | 0.554 | 0.522 | 0.421 | 0.415 | 0.433 | 0.432 |
| | Avg | 0.279 | 0.314 | 0.288 | 0.332 | 0.287 | 0.333 | 0.293 | 0.347 | 0.288 | 0.332 | 0.286 | 0.327 | 0.281 | 0.326 | 0.757 | 0.610 | 0.358 | 0.404 | 0.291 | 0.333 | 0.350 | 0.401 | 0.305 | 0.349 | 0.327 | 0.371 |
| ETTh1 | 96 | **0.374** | **0.393** | 0.386 | 0.405 | 0.390 | 0.403 | 0.398 | 0.416 | 0.386 | 0.405 | 0.386 | 0.395 | 0.414 | 0.419 | 0.423 | 0.448 | 0.479 | 0.464 | 0.384 | 0.402 | 0.386 | 0.400 | 0.376 | 0.419 | 0.449 | 0.459 |
| | 192 | **0.427** | **0.422** | 0.443 | 0.437 | 0.451 | 0.438 | 0.451 | 0.446 | 0.441 | 0.436 | 0.437 | 0.424 | 0.460 | 0.445 | 0.471 | 0.474 | 0.525 | 0.492 | 0.436 | 0.429 | 0.437 | 0.432 | 0.420 | 0.448 | 0.500 | 0.482 |
| | 336 | **0.447** | **0.441** | 0.489 | 0.468 | 0.496 | 0.458 | 0.497 | 0.473 | 0.487 | 0.458 | 0.479 | 0.446 | 0.501 | 0.466 | 0.570 | 0.546 | 0.565 | 0.515 | 0.491 | 0.469 | 0.481 | 0.459 | 0.459 | 0.465 | 0.521 | 0.496 |
| | 720 | **0.469** | **0.463** | 0.502 | 0.489 | 0.520 | 0.493 | 0.526 | 0.509 | 0.503 | 0.491 | 0.481 | 0.470 | 0.500 | 0.488 | 0.653 | 0.621 | 0.594 | 0.558 | 0.521 | 0.500 | 0.519 | 0.516 | 0.506 | 0.507 | 0.514 | 0.512 |
| | Avg | **0.434** | **0.430** | 0.455 | 0.450 | 0.439 | 0.448 | 0.468 | 0.461 | 0.454 | 0.447 | 0.446 | 0.434 | 0.469 | 0.454 | 0.529 | 0.522 | 0.541 | 0.507 | 0.458 | 0.450 | 0.456 | 0.452 | 0.440 | 0.460 | 0.496 | 0.487 |
| ETTh2 | 96 | **0.287** | **0.337** | 0.296 | 0.348 | 0.298 | 0.351 | 0.307 | 0.363 | 0.297 | 0.349 | 0.288 | 0.338 | 0.302 | 0.348 | 0.745 | 0.584 | 0.400 | 0.440 | 0.340 | 0.374 | 0.333 | 0.387 | 0.358 | 0.397 | 0.346 | 0.388 |
| | 192 | **0.370** | **0.388** | 0.376 | 0.396 | 0.393 | 0.407 | 0.394 | 0.414 | 0.380 | 0.400 | 0.374 | 0.390 | 0.388 | 0.400 | 0.877 | 0.656 | 0.528 | 0.509 | 0.402 | 0.414 | 0.477 | 0.476 | 0.429 | 0.439 | 0.456 | 0.452 |
| | 336 | **0.412** | **0.425** | 0.424 | 0.431 | 0.436 | 0.441 | 0.437 | 0.447 | 0.428 | 0.432 | 0.415 | 0.426 | 0.426 | 0.433 | 1.043 | 0.731 | 0.643 | 0.571 | 0.452 | 0.452 | 0.594 | 0.541 | 0.496 | 0.487 | 0.482 | 0.486 |
| | 720 | **0.419** | **0.438** | 0.426 | 0.444 | 0.431 | 0.449 | 0.445 | 0.462 | 0.427 | 0.445 | 0.420 | 0.440 | 0.431 | 0.446 | 1.104 | 0.763 | 0.874 | 0.679 | 0.462 | 0.468 | 0.831 | 0.657 | 0.463 | 0.474 | 0.515 | 0.511 |
| | Avg | **0.372** | **0.396** | 0.381 | 0.405 | 0.390 | 0.412 | 0.396 | 0.422 | 0.383 | 0.407 | 0.374 | 0.398 | 0.387 | 0.407 | 0.942 | 0.684 | 0.611 | 0.550 | 0.414 | 0.427 | 0.559 | 0.515 | 0.437 | 0.449 | 0.450 | 0.459 |
| Electricity | 96 | **0.137** | **0.234** | 0.139 | 0.235 | 0.192 | 0.280 | 0.146 | 0.246 | 0.148 | 0.240 | 0.201 | 0.281 | 0.181 | 0.270 | 0.219 | 0.314 | 0.237 | 0.329 | 0.168 | 0.272 | 0.197 | 0.282 | 0.193 | 0.308 | 0.201 | 0.317 |
| | 192 | **0.158** | 0.251 | 0.159 | 0.255 | 0.191 | 0.280 | 0.167 | 0.265 | 0.162 | 0.253 | 0.201 | 0.283 | 0.188 | 0.274 | 0.231 | 0.322 | 0.236 | 0.330 | 0.184 | 0.289 | 0.196 | 0.285 | 0.201 | 0.315 | 0.222 | 0.334 |
| | 336 | 0.182 | **0.173** | 0.176 | 0.272 | 0.211 | 0.299 | 0.182 | 0.281 | 0.178 | 0.269 | 0.215 | 0.298 | 0.204 | 0.293 | 0.246 | 0.337 | 0.249 | 0.344 | 0.198 | 0.300 | 0.209 | 0.301 | 0.214 | 0.329 | 0.231 | 0.338 |
| | 720 | **0.200** | **0.292** | 0.204 | 0.298 | 0.264 | 0.340 | 0.208 | 0.304 | 0.225 | 0.317 | 0.257 | 0.331 | 0.246 | 0.324 | 0.280 | 0.363 | 0.284 | 0.373 | 0.220 | 0.320 | 0.245 | 0.333 | 0.246 | 0.355 | 0.254 | 0.361 |
| | Avg | **0.170** | **0.238** | 0.170 | 0.265 | 0.215 | 0.300 | 0.176 | 0.274 | 0.178 | 0.270 | 0.219 | 0.298 | 0.205 | 0.290 | 0.244 | 0.334 | 0.251 | 0.344 | 0.192 | 0.295 | 0.212 | 0.300 | 0.214 | 0.327 | 0.227 | 0.338 |
| Exchange | 96 | **0.085** | **0.205** | 0.086 | 0.207 | 0.091 | 0.216 | 0.103 | 0.233 | 0.086 | 0.206 | 0.093 | 0.217 | 0.088 | 0.205 | 0.256 | 0.367 | 0.094 | 0.218 | 0.107 | 0.234 | 0.088 | 0.218 | 0.148 | 0.278 | 0.197 | 0.323 |
| | 192 | **0.175** | 0.297 | 0.182 | 0.304 | 0.189 | 0.313 | 0.214 | 0.337 | 0.177 | 0.299 | 0.184 | 0.307 | 0.176 | 0.299 | 0.470 | 0.509 | 0.184 | 0.307 | 0.226 | 0.344 | 0.176 | 0.315 | 0.271 | 0.315 | 0.300 | 0.369 |
| | 336 | 0.317 | **0.407** | 0.332 | 0.418 | 0.333 | 0.421 | 0.366 | 0.445 | 0.331 | 0.417 | 0.351 | 0.432 | 0.301 | 0.397 | 1.268 | 0.883 | 0.349 | 0.431 | 0.367 | 0.448 | 0.313 | 0.427 | 0.460 | 0.427 | 0.509 | 0.524 |
| | 720 | **0.825** | 0.683 | 0.867 | 0.703 | 0.916 | 0.729 | 0.931 | 0.738 | 0.847 | 0.691 | 0.886 | 0.714 | 0.901 | 0.714 | 1.767 | 1.068 | 0.852 | 0.698 | 0.964 | 0.746 | 0.839 | 0.695 | 1.195 | 0.695 | 1.447 | 0.941 |
| | Avg | **0.351** | **0.400** | 0.367 | 0.408 | 0.382 | 0.420 | 0.404 | 0.428 | 0.360 | 0.403 | 0.378 | 0.417 | 0.367 | 0.404 | 0.940 | 0.707 | 0.370 | 0.413 | 0.416 | 0.443 | 0.354 | 0.414 | 0.519 | 0.429 | 0.613 | 0.539 |
| Solar-Energy | 96 | **0.202** | **0.233** | 0.205 | 0.244 | 0.238 | 0.277 | 0.231 | 0.286 | 0.203 | 0.237 | 0.322 | 0.339 | 0.234 | 0.286 | 0.310 | 0.331 | 0.312 | 0.399 | 0.250 | 0.292 | 0.290 | 0.378 | 0.242 | 0.342 | 0.884 | 0.711 |
| | 192 | **0.230** | **0.254** | 0.237 | 0.270 | 0.299 | 0.319 | 0.257 | 0.285 | 0.233 | 0.261 | 0.359 | 0.356 | 0.267 | 0.310 | 0.734 | 0.725 | 0.339 | 0.416 | 0.296 | 0.318 | 0.320 | 0.398 | 0.285 | 0.380 | 0.834 | 0.692 |
| | 336 | 0.254 | **0.265** | 0.258 | 0.288 | 0.310 | 0.327 | 0.256 | 0.293 | 0.248 | 0.273 | 0.397 | 0.369 | 0.290 | 0.315 | 0.750 | 0.735 | 0.368 | 0.430 | 0.319 | 0.330 | 0.353 | 0.415 | 0.282 | 0.376 | 0.941 | 0.723 |
| | 720 | 0.252 | **0.271** | 0.260 | 0.288 | 0.310 | 0.330 | 0.252 | 0.295 | 0.249 | 0.275 | 0.397 | 0.356 | 0.289 | 0.317 | 0.769 | 0.765 | 0.370 | 0.425 | 0.338 | 0.337 | 0.356 | 0.413 | 0.357 | 0.427 | 0.882 | 0.717 |
| | Avg | **0.235** | **0.256** | 0.240 | 0.273 | 0.289 | 0.313 | 0.249 | 0.290 | 0.233 | 0.262 | 0.369 | 0.356 | 0.270 | 0.307 | 0.641 | 0.639 | 0.347 | 0.417 | 0.301 | 0.319 | 0.330 | 0.401 | 0.291 | 0.381 | 0.885 | 0.711 |
| PEMS04 | 12 | **0.075** | 0.182 | 0.076 | 0.180 | 0.110 | 0.226 | 0.082 | 0.193 | 0.078 | 0.183 | 0.138 | 0.252 | 0.105 | 0.224 | 0.098 | 0.218 | 0.219 | 0.340 | 0.087 | 0.195 | 0.148 | 0.272 | 0.138 | 0.262 | 0.424 | 0.491 |
| | 24 | **0.084** | **0.193** | 0.084 | 0.193 | 0.161 | 0.275 | 0.099 | 0.214 | 0.095 | 0.205 | 0.258 | 0.348 | 0.153 | 0.205 | 0.131 | 0.256 | 0.292 | 0.398 | 0.103 | 0.215 | 0.224 | 0.340 | 0.177 | 0.293 | 0.459 | 0.509 |
| | 48 | **0.105** | **0.217** | 0.115 | 0.224 | 0.345 | 0.403 | 0.123 | 0.240 | 0.120 | 0.233 | 0.572 | 0.544 | 0.229 | 0.339 | 0.205 | 0.326 | 0.409 | 0.478 | 0.136 | 0.250 | 0.355 | 0.437 | 0.270 | 0.368 | 0.646 | 0.610 |
| | 96 | **0.130** | **0.243** | 0.137 | 0.248 | 0.588 | 0.553 | 0.151 | 0.267 | 0.150 | 0.262 | 1.137 | 0.820 | 0.291 | 0.389 | 0.402 | 0.457 | 0.492 | 0.532 | 0.190 | 0.303 | 0.452 | 0.504 | 0.341 | 0.427 | 0.912 | 0.748 |
| | Avg | **0.099** | **0.209** | 0.103 | 0.211 | 0.301 | 0.364 | 0.114 | 0.229 | 0.111 | 0.221 | 0.526 | 0.491 | 0.195 | 0.307 | 0.209 | 0.314 | 0.353 | 0.437 | 0.129 | 0.241 | 0.295 | 0.388 | 0.231 | 0.337 | 0.610 | 0.590 |
| PEMS08 | 12 | **0.075** | **0.177** | 0.076 | 0.178 | 0.099 | 0.214 | 0.080 | 0.190 | 0.079 | 0.182 | 0.133 | 0.247 | 0.168 | 0.232 | 0.165 | 0.214 | 0.227 | 0.343 | 0.112 | 0.212 | 0.154 | 0.276 | 0.173 | 0.273 | 0.436 | 0.485 |
| | 24 | **0.102** | **0.207** | 0.104 | 0.209 | 0.169 | 0.277 | 0.114 | 0.223 | 0.115 | 0.219 | 0.249 | 0.343 | 0.224 | 0.281 | 0.215 | 0.260 | 0.318 | 0.409 | 0.141 | 0.238 | 0.248 | 0.353 | 0.210 | 0.301 | 0.467 | 0.502 |
| | 48 | **0.154** | **0.226** | 0.167 | 0.228 | 0.274 | 0.360 | 0.175 | 0.271 | 0.186 | 0.235 | 0.569 | 0.544 | 0.321 | 0.354 | 0.315 | 0.355 | 0.497 | 0.510 | 0.198 | 0.283 | 0.440 | 0.470 | 0.320 | 0.394 | 0.966 | 0.733 |
| | 96 | 0.243 | 0.305 | 0.245 | 0.280 | 0.522 | 0.499 | 0.298 | 0.348 | **0.221** | **0.267** | 1.166 | 0.814 | 0.408 | 0.417 | 0.377 | 0.397 | 0.721 | 0.592 | 0.320 | 0.351 | 0.674 | 0.565 | 0.442 | 0.465 | 1.385 | 0.915 |
| | Avg | **0.145** | 0.228 | 0.148 | 0.223 | 0.266 | 0.338 | 0.167 | 0.258 | 0.150 | 0.226 | 0.529 | 0.487 | 0.280 | 0.321 | 0.268 | 0.307 | 0.441 | 0.464 | 0.193 | 0.271 | 0.379 | 0.416 | 0.286 | 0.358 | 0.814 | 0.659 |

performance of FLDmamba compared to S-Mamba directly demonstrates the efficacy of integrating Fourier and Laplace transforms within our framework, leading to enhanced resilience against noise. Conversely, the iTransformer model exhibits the most substantial performance degradation in these robustness tests, indicating a lower tolerance to noisy input data.

## 4.5 LONG-TERM PREDICTION COMPARISON (Q4)

This section investigates the effectiveness of our proposed framework, FLDmamba, in long-term time series prediction compared to other state-of-the-art methods. We conduct a comparative analysis against Transformer-based baselines (iTransformer, Rlinear, Autoformer) and a related Mamba-based method (S-Mamba). The results, presented in Figure 5 and Figure 11 (Appendix 6.5), reveal the following key observations: **Superior Long-Term Performance of Mamba-Based Methods:** Compared to Transformer-based baselines, both S-Mamba and our method, FLDmamba, which are based on the Mamba architecture, demonstrate

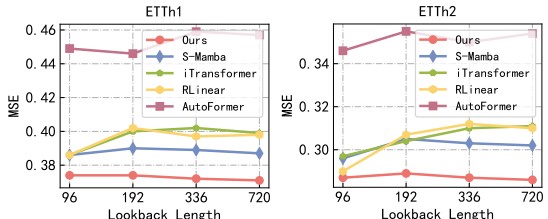

Figure 5: Long-term prediction with the lookback length from the range [96, 192, 336, 720].

superior performance in terms of MAE and MSE. Furthermore, both methods exhibit a reduced or stable performance trend as the lookback window size increases from 96 to 720. This indicates that Mamba-based methods are more adept at capturing temporal patterns and dependencies, effectively preserving sequential features in long-term time series data. **Enhanced Long-Term Prediction with FLDmamba:** Comparing FLDmamba to S-Mamba, our method shows a clear trend of reduced or maintained performance with increasing lookback window size. We attribute this improvement to the incorporation of the Fourier transform and the inverse Laplace transform, which

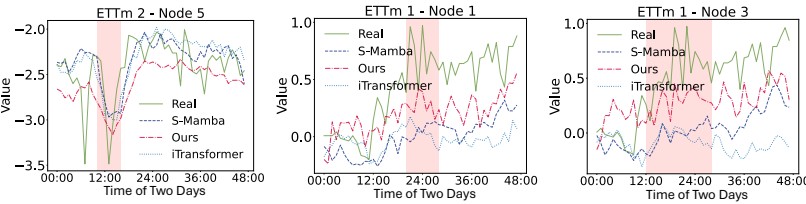

Figure 7: Case study of FLDmamba in terms of transient dynamics like short-term fluctuations.

effectively capture periodic dependencies and further enhance the ability to handle long-term prediction. These findings highlight the effectiveness of FLDmamba in capturing complex temporal dynamics and maintaining performance even with extended lookback windows, demonstrating its significant advantage for long-term time series prediction.

### 4.6 CASE STUDY (Q5)

This section examines the efficacy of our proposed framework, FLDmamba, in capturing multi-scale periodicity and transient dynamics, particularly short-term fluctuations, within time series data. To illustrate its capabilities, we present case studies based on the ETTm1 and ETTm2 datasets, as depicted in Figure 6, Figure 13 (Appendix 6.5) and Figure 7. These figures showcase the variations in the datasets over two consecutive days and 12 hours, respectively. For comparative analysis, we include the predicted results of two state-of-the-art baselines, S-Mamba and iTransformer. Each plot displays four curves: the ground truth values, the predictions generated by S-Mamba, the predictions from iTransformer, and the predictions obtained using our FLDmamba. We have the following observations: **Enhanced Multi-Scale Periodicity Capture:**
As illustrated in Figure 6 and Figure 13 (Appendix 6.5), our proposed framework, FLDmamba, demonstrates a distinct advantage in capturing multi-scale periodicity within time series data when compared to both S-Mamba and iTransformer. This enhanced ability to model periodic patterns, which are often characteristic of time series data, contributes significantly to its improved accuracy in time series prediction, further validating the effectiveness of our approach. Notably, the comparison with S-Mamba predictions reinforces the significant contribution of the Fourier Transform and Laplace Transform in capturing multi-scale periodicity and, subsequently, improving prediction performance. The inclusion of these transforms within our framework allows for a more comprehensive and nuanced understanding of the underlying periodic patterns

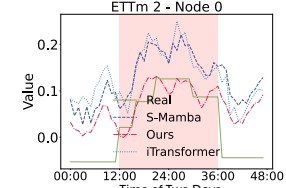

Figure 6: Case study of FLDmamba in terms of multi-scale periodicity.

present in the data, leading to more accurate predictions. **Improved Transient Dynamics Capture:** Figure 7 showcases the effectiveness of our method, FLDmamba, in capturing transient dynamics, particularly short-term fluctuations, compared to S-Mamba and iTransformer. The Laplace Transform within our framework significantly enhances its ability to model these dynamics, leading to improved performance. While S-Mamba demonstrates some capability in capturing transient dynamics, our method exhibits a more pronounced advantage. Conversely, iTransformer shows limited effectiveness in capturing these short-term fluctuations.

### 5 CONCLUSION

In conclusion, this paper addresses the limitations of existing time series prediction models, particularly in capturing multi-scale periodicity, transient dynamics and noise alleviation within long-term predictions. We propose a novel framework, FLDmamba, which leverages the strengths of both Fourier and Laplace transforms to effectively address these challenges. By integrating Fourier analysis into Mamba, FLDmamba enhances its ability to capture global-scale properties, such as multi-scale patterns, in the frequency domain. Our extensive experiments demonstrate that FLDmamba achieves state-of-the-art performance in most of cases on 9 datasets on time series prediction benchmarks. This work offers a effective and robust solution for long-term time series prediction, paving the way for its application in real-world scenarios. Future investigations will further enhance the model's adaptability to dynamic data environments. Detailed limitations and future work discussion of our paper are shown in Appendix 6.8.

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

# 6 APPENDIX

## 6.1 ALGORITHMS

---

**Algorithm 1:** The **FLDmamba** Algorithm

---

**Input:** $\mathbf{X}$: (B, L, V);
**Output:** $\widehat{\mathbf{Y}}$: (B, L, V);

1   $U \leftarrow$ **FMamba**($\mathbf{X}$);    // Step into FMamba algorithm 2
2   $U' \leftarrow$ **Mamba**($\mathbf{X}$);    // Step into the Mamba algorithm 3
3   $U'' \leftarrow U' + U$;
4   $y' \leftarrow$ **FMamba**($U''$);    // Step into FMamba algorithm 2;
5   $y'' \leftarrow$ **Mamba**($U''$);    // Step into Mamba algorithm 3;
6   $Y \leftarrow$ FFT($y' + y''$);
7   $Y \leftarrow$ Linear($Y$);
8   $\widehat{\mathbf{Y}} \leftarrow$ ILT($Y$);    // Inverse Laplace Transform module
9   **return** $\widehat{\mathbf{Y}}$;

---

**Algorithm 2:** The **FMamba** Algorithm

---

**Input:** $\mathbf{X}$: (B, L, V);
**Output:** $U$:(B, L, V);

1   $\mathbf{X}' \leftarrow$ RBF($\mathbf{X}$);
2   **for** $p = 1, 2, ..., FMamba\ layers$ **do**
3      $\mathbf{A}$: (V, N) $\leftarrow$ Parameter
4      $\mathbf{B}$: (V, L, N) $\leftarrow s_B(\mathbf{X}')$
5      $\mathbf{C}$: (B, L, N) $\leftarrow s_C(\mathbf{X}')$
6      $\Delta$: (B, L, N) $\leftarrow \tau_\Delta$(Parameter $+ s_\Delta(\mathbf{X}')$)
7      $\Delta' =$ FFT($\Delta$)
8      $\Delta_F =$ IFFT($\tilde{W} \cdot \Delta'$)
9      $\bar{\mathbf{A}}_F, \bar{\mathbf{B}}_F$ : (B, L, V, N) $\leftarrow discretize(\Delta_F, \mathbf{A}, \mathbf{B})$
10     $U^{(1)} \leftarrow$ SSM ($\bar{\mathbf{A}}_F, \bar{\mathbf{B}}_F, \mathbf{C}$)($\mathbf{X}'$)
11     $U^{(2)} \leftarrow U^{(1)} \otimes$ SiLU($Linear(\mathbf{X}')$)
12     $U \leftarrow Linear(U^{(2)})$
13   **end**
14   **return** $U$;

---

**Algorithm 3:** The **Mamba** Algorithm

---

**Input:** $\mathbf{X}$: (B, L, V);
**Output:** $U'$:(B, L, V);

1   $\mathbf{X}' \leftarrow$ RBF($\mathbf{X}$);
2   **for** $p = 1, 2, ..., Mamba\ layers$ **do**
3      $\mathbf{A}$: (V, N) $\leftarrow$ Parameter
4      $\mathbf{B}$: (B, L, N) $\leftarrow s_B(\mathbf{X}')$
5      $\mathbf{C}$: (B, L, N) $\leftarrow s_C(\mathbf{X}')$
6      $\Delta$: (B, L, N) $\leftarrow \tau_\Delta$(Parameter $+ s_\Delta(\mathbf{X}')$)
7      $\bar{\mathbf{A}}, \bar{\mathbf{B}}$ : (B, L, V, N) $\leftarrow discretize(\Delta, \mathbf{A}, \mathbf{B})$
8      $U'^{(1)} \leftarrow$ SSM ($\bar{\mathbf{A}}, \bar{\mathbf{B}}, \mathbf{C}$)($\mathbf{X}'$)
9      $U'^{(2)} \leftarrow U'^{(1)} \otimes$ SiLU($Linear(\mathbf{X}')$)
10     $U' \leftarrow Linear(U'^{(2)})$
11   **end**
12   **return** $U'$;

---

## 6.2 PRELIMINARY

**Mamba**. Mamba is proposed in (Gu & Dao, 2023). With four parameters $\mathbf{A}, \mathbf{B}, \mathbf{C}, \Delta$, Mamba is defined based on a sequence-to-sequence transformation via the following equations:

$$h'(t) = \mathbf{A}h(t) + \mathbf{B}x(t);$$
$$y(t) = \mathbf{C}h(t);$$
$$h_t = \bar{\mathbf{A}}h_{t-1} + \bar{\mathbf{B}}x_t \tag{11}$$

where $h(t)$ denotes the hidden state, $x(t)$ is the input sequence, $y(t)$ is the output sequence, and $\mathbf{A} \in \mathbb{R}^{N \times N}, \mathbf{B} \in \mathbb{R}^{L \times N}, \mathbf{C} \in \mathbb{R}^{N \times L}$. In addition, $N$ and $L$ are the dimension factor and the sequence length, respectively. The discretization process of parameters $(A, B)$ is shown as follows:

$$\bar{\mathbf{A}} = \exp(\Delta\mathbf{A}); \quad \bar{\mathbf{B}} = \Delta\mathbf{A}^{-1}\exp(\Delta\mathbf{A}) \cdot \Delta\mathbf{B} \tag{12}$$

Here the discretization is closely related to continuous-time systems, providing them with additional properties such as resolution invariance (Nguyen et al., 2022) and automatic normalization, ensuring the model's proper calibration. Mamba achieves input-dependent selection by making $\mathbf{B}$, $\mathbf{C}$, and $\mathbf{\Delta}$ functions of the input $x$. In this way, Mamba is able to dynamically adjust its operations, computations, and information flow based on the specific characteristics of the input data. This input-dependent selection allows Mamba to effectively adapt its behavior and capture the relevant patterns and dynamics present in the input, resulting in enhanced modeling capabilities and improved performance for various tasks. Then a state-space model (SSM) utilize $\bar{\mathbf{A}}, \bar{\mathbf{B}}$, and $\mathbf{C}$ to process the input $x$:

$$\bar{\mathbf{K}} = (\mathbf{C}\bar{\mathbf{B}}, \mathbf{C}\bar{\mathbf{A}}\bar{\mathbf{B}}, ...\mathbf{C}\bar{\mathbf{A}}^k\bar{\mathbf{B}}, ...)^T, \quad y = \bar{\mathbf{K}}^T x \tag{13}$$

Finally, the output $y$ of the SSM is multiplied with a non-linear activation-transformed input. This result is then passed through a final linear layer to produce Mamba's output. For a complete overview of Mamba's architecture, refer to Algorithm 3.

**Fourier Transform**. Given the input function $\mathbf{f}(x)$, we can obtain the frequency domain conversion function $\mathbf{F}(k)$ via the Discrete Fourier Transform (DFT), where $\mathbf{F}$ denotes the Fourier transform of the function $\mathbf{f}(x)$. The process is shown as follows:

$$\mathbf{F}(k) = \int_d \mathbf{f}(x)e^{-j2\pi kx}dx$$
$$= \int_d \mathbf{f}(x)\cos(2\pi kx)dx + j\int_d \mathbf{f}(x)\sin(2\pi kx)dx \tag{14}$$

In this context, we have the frequency variable denoted as $k$, the spatial variable as $x$, and the imaginary unit as $j$. The real part of $\mathbf{F}$ is represented as $\text{Re}(\mathbf{F})$, while the imaginary part is denoted as $\text{Im}(\mathbf{F})$. The complete conversion is expressed as $\mathbf{F} = \text{Re}(\mathbf{F}) + j\text{Im}(\mathbf{F})$. The Fourier transform is employed to decompose the input signal into its constituent frequencies. This process facilitates the identification and detection of periodic or aperiodic patterns, which are crucial for tasks such as image recognition.

**Laplace Analysis**. The Laplace analysis is a powerful mathematical tool used in various fields, particularly in engineering, physics, and applied mathematics. It allows us to convert functions of time into functions of complex variables, providing a useful way to analyze and solve differential equations. Below, we provide preliminaries for the Laplace analysis, and also show how our inverse Laplace transform can capture transient dynamics.

The Laplace transform of a function, denoted as $F(s)$, is defined as follows:

$$F(s) = \mathcal{L}\{f(t)\} = \int_0^\infty e^{-st}f(t)\,dt \tag{15}$$

In this equation, $f(t)$ is the original function in the time domain, $s$ is a complex variable, and $F(s)$ is the transformed function in the complex frequency domain. The Laplace transform has several important properties that make it a versatile tool for analysis. For example, it enables us to simplify differential equations into algebraic equations, making it easier to solve for unknown

functions. Additionally, the Laplace transform allows us to study system behavior, stability, and response to different inputs. By applying the inverse Laplace transform, we can obtain the original function back from its transformed representation. This transformation provides a valuable method for understanding and manipulating functions in the frequency domain, facilitating analysis and design in various scientific and engineering disciplines.

The inverse Laplace transform is defined as follows:

$$f(t) = \mathcal{L}^{-1}\{F(s)\} = \lim_{T \to \infty} \int_{\gamma-iT}^{\gamma+iT} e^{st} F(s) \, ds \tag{16}$$

Here $\text{Re}(s) = \gamma$ and $\gamma$ is greater than the real part of all singularities of $F(s)$. For general functions, the inverse Laplace transform may not have analytical solution.

To allow analytical solution for inverse Laplace transform, we follow (Cao et al., 2023) and consider a neural operator which maps a function $v(t)$ to the function $u(t)$:

$$u(t) = (\kappa(\phi) * v)(t) = \int_D \kappa_\phi(t - \tau) v(\tau) d\tau \tag{17}$$

where $\kappa$ is a kernel integral transformation. Imposing $\kappa_\phi(t, \tau) = \kappa_\phi(t - \tau)$, in the Laplace space we have

$$U(s) = K_\phi(s) V(s) \tag{18}$$

where $K_\phi(s) = \mathcal{L}\{\kappa_\phi(t)\}$ and $V(s) = \mathcal{L}\{v(t)\}, U(s) = \mathcal{L}\{u(t)\}$.

Here we assume that the kernel integral operator has the form of $K_\phi(s) = \sum_{n=1}^{N} \frac{\beta_n}{s - \mu_n}$ in the Laplace space, where $\beta_n \in \mathbb{R}$ and $\mu_n \in \mathbb{C}$ are learnable parameters. Also, performing Fourier transform on $v(t)$, we have $v(t) = \sum_{l=-\infty}^{\infty} \alpha_l \exp i\omega_l t$, which results in $V(s) = \sum_{l=-\infty}^{\infty} \frac{\alpha_l}{s - i\omega_l}$. We make the assumption so that the singularities are first-order, and the inverse Laplace transform has analytical solution. After some derivation, we have that the resulting form for $u(t)$ in the original space is

$$u(t) = \sum_{n=1}^{N} \gamma_n \exp(\mu_n t) + \sum_{l=-\infty}^{\infty} \lambda_l \exp(i\omega_l t) \tag{19}$$

Here $\omega_l$ are frequencies by decomposing $v(t)$ via Fourier series, and $\gamma_n, \lambda_l$ are derived parameters from $\beta_n, \omega_l$ and $\mu_n$. For detailed derivation, see (Cao et al., 2023). If we truncate the number of Fourier series terms $l$, the above Eq. 19 reduces to

**R#6yv6-W1**

$$u(t) = \sum_{n=1}^{M} A_n e^{-\sigma_n t} \cos(w_n t + \varphi_n) \tag{20}$$

In our work, we directly parameterize the above $A_n$, $\sigma_n$, $w_n$, and $\varphi_n$ as learnable functions of the output of the previous layer, which in turn are functions of the history time series.

Here we see that equation 20 exactly describes transient dynamics, characterized by decay rate $\sigma_n$ and periods $w_n$. Therefore, our parameterization of the inverse Laplace transform via Eq. 20 can learn transient dynamics accurately.

Furthermore, in contrast to performing inverse Laplace transform which involves integration in the complex plane where the integrand has poles, we see that our parameterization in Eq. 20 has better efficiency and stability.

## 6.3 MORE RELATED WORK

While several Mamba-based methods exist for time series prediction, such as those by Wang et al. ( Wang et al. (2024)), Xu et al. ( Xu et al.), and Liang et al. ( Liang et al. (2024)), our approach distinctly differs in its focus and methodology. For instance, Wang et al. (2024) independently tokenize time points for each variable using a linear layer, employ a bidirectional Mamba layer to capture inter-variable correlations, and utilize a Feed-Forward network for learning temporal dependencies, ultimately producing forecasts through a linear mapping layer. In contrast, Xu et al. ( Xu et al. leverage Mamba to identify global patterns in coarse-grained long-range time series, while the Local Window Transformer (LWT) focuses on local variations in fine-grained short-range time series. Liang et al. (2024) introduce a patching technique aimed at enhancing local information and capturing evolving patterns to address sparse time series semantics, primarily targeting long-term predictions with high efficiency. In contrast to these studies, our method not only handles long-term prediction efficiency but also emphasizes capturing multi-scale periodicity and addressing transient dynamics through the integration of Fourier and Laplace transforms. By incorporating these transforms, we also tackle the issue of data noise in time series, setting our approach apart from existing methods that primarily focus on long-term prediction efficiency.

## 6.4 MODEL COMPLEXITY

This section presents a complexity analysis of our proposed model, FLDmamba. The computational complexity of the base Mamba model is $\mathcal{O}(BLVN)$, where $B$ represents the batch size, $L$ denotes the sequence length, $V$ signifies the number of variables, and $N$ indicates the state expansion factor. The Fast Fourier Transform (FFT) in FLDmamba has time complexity of $\mathcal{O}(BLN \log L)$, and the inverse Laplace transform has time complexity of $\mathcal{O}(BLN)$, both significantly smaller than $\mathcal{O}(BLVN)$. Therefore, the total time complexity is still $\mathcal{O}(BLVN)$. In other words, FLDmamba maintains a comparable computational time complexity to the base Mamba model, making it a promising framework for large-scale real-world applications in time series prediction. This computational efficiency allows FLDmamba to handle extensive datasets and complex time series scenarios without significant performance degradation.

## 6.5 EXPERIMENTS

### 6.5.1 EXPERIMENT SETTINGS

To ensure a fair comparison, we modify the hidden dimensionality of all compared algorithms within the range of $[128, 256, 512, 1024, 2048]$ to achieve their reported best performance, which is consistently observed at 1024. The learning rate ($\eta$) is initialized to $5 \times 10^{-6}$, and we set the number of FLDmamba layers to 2. Consistent with the existing settings of time series datasets, we utilize historical data with 96, 192, 336, or 720 time steps. The time steps are defined as 5 minutes, 1 hour, 10 minutes, or 1 day intervals to predict the corresponding future 96, 192, 336, or 720 time steps in these time series datasets. All baseline methods are evaluated using their predefined settings as described in their respective publications. We conduct testing for all tasks on a single NVIDIA L40 GPU equipped with 128 CPUs.

Table 2: The statistics of 9 public datasets.

| Datasets | Variates | Timesteps | Granularity |
|---|---|---|---|
| ETTh1&ETTh2 | 7 | 69,680 | 1 hour |
| PEMS04 | 307 | 16,992 | 5 minutes |
| PEMS08 | 170 | 17,856 | 5 minutes |
| Exchange | 8 | 7,588 | 1 day |
| Electricity | 321 | 26,304 | 1 hour |
| Solar-Energy | 137 | 52,560 | 10 minutes |
| ETTm1&ETTm2 | 7 | 17,420 | 15min |

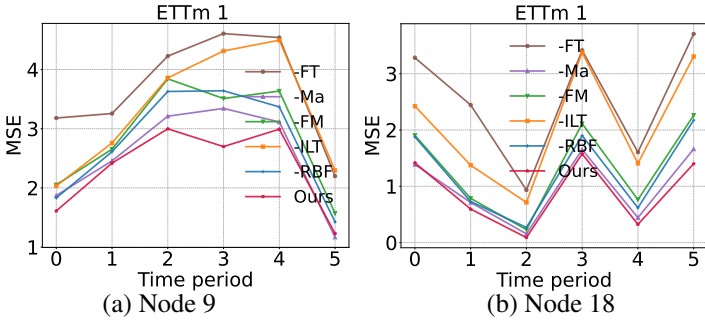

Figure 8: Ablation study of FLDmamba on prediction performance on Node 9 and Node 18 instances of ETTm 1 dataset.

### 6.5.2 BASELINE DESCRIPTIONS

Transformer-based methods:

- Autoformer (Wu et al., 2021) employs a series decomposition technique along with an Auto-Correlation mechanism to effectively capture cross-time dependencies.
- FEDformer (Zhou et al., 2022) introduces an enhanced Transformer operating in the frequency domain, aiming to improve both efficiency and effectiveness.
- Crossformer (Zhang & Yan, 2022) incorporates a patching operation like other models but distinguishes itself by employing Cross-Dimension attention to capture dependencies between different series. While patching reduces the elements to process and extracts semantic information comprehensively, these models encounter performance limitations when handling longer.
- DLinear (Zeng et al., 2023) introduced DLinear, a method that decomposes time series into two distinct components and generates a single Linear layer for each component. This straightforward design has outperformed all previously proposed complex transformer models.
- PatchTST (Huang et al., 2024) leverages patching and channel-independent techniques to facilitate the extraction of semantic information from single time steps to multiple time steps within time series data.
- iTransFormer (Liu et al., 2023) employs inverted attention layers to effectively capture inter-series dependencies. However, its tokenization approach, which involves passing the entire sequence through a Multilayer Perceptron (MLP) layer, falls short in capturing the complex evolutionary patterns inherent in time series data.

MLP-based methods:

- TimesNet (Wu et al., 2022) expands the examination of temporal fluctuations by extending the 1-D time series into a collection of 2-D tensors across multiple periods.
- RLinear (Li et al., 2023), the state-of-the-art linear model, incorporates reversible normalization and channel independence into a purely linear structure.
- TiDE (Das et al., 2023) is an encoder-decoder model that employs a Multi-layer Perceptron (MLP) architecture.

SSM-based methods:

- S-Mamba (Wang et al., 2024) independently tokenizes the time points for each variate using a linear layer. This allows for the extraction of correlations between variates using a bidirectional Mamba layer, while a Feed-Forward Network is employed to learn temporal dependencies. **R#vEmK-W5, R#RZwJ-W1, R#6yv6-W2**
- SST (Xu et al.) leverages Mamba to identify global patterns in coarse-grained long-range time series, while the Local Window Transformer (LWT) focuses on local variations in fine-grained short-range time series.

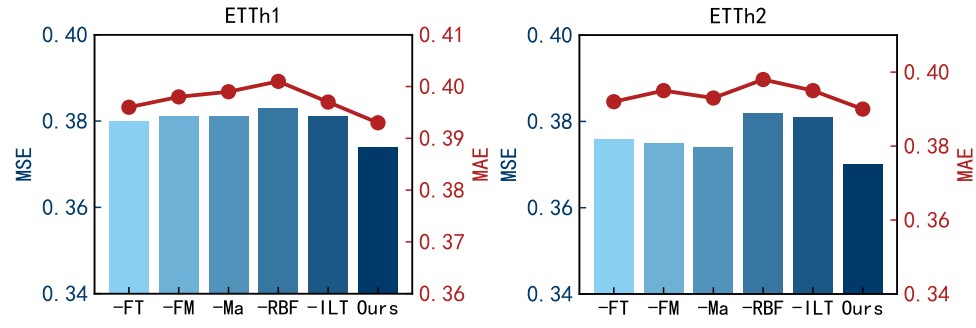

Figure 9: Ablation study of FLDmamba on four datasets with $L = 96$.

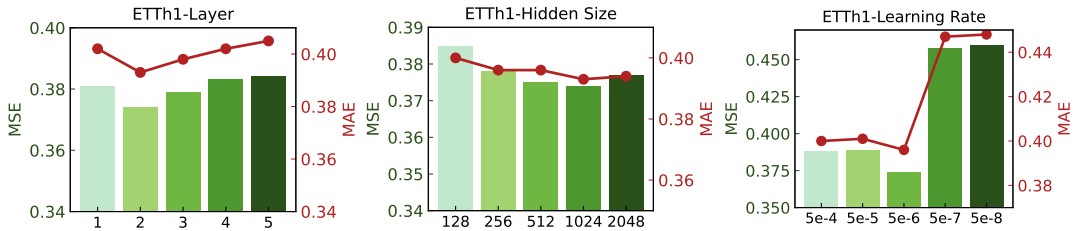

Figure 10: Hyperparameter study of FLDmamba.

- Bi-mamba+ Liang et al. (2024) introduces a patching technique aimed at enhancing local information and capturing evolving patterns to address sparse time series semantics, primarily targeting long-term predictions with high efficiency.

### 6.6 PEARSON CORRELATION

We also calculated Pearson correlation and show results in Table 3. The results indicate that our method consistently outperforms other baselines across most cases and all datasets, further confirming its superior performance.

R#RZwJ-W1

Table 3: Performance comparison in terms of Pearson correlation

R#RZwJ-W1

| Models | Metric | ETTm1 | ETTm2 | ETTh1 | ETTh2 | Electricity | Exchange | Solar-Energy | Metric | PEMS04 | PEMS08 |
|---|---|---|---|---|---|---|---|---|---|---|---|
| FLDmamba (**ours**) | 96 | **0.857** | **0.950** | **0.892** | **0.920** | 0.929 | **0.978** | **0.818** | 12 | **0.793** | **0.839** |
| | 192 | **0.830** | **0.935** | 0.799 | **0.898** | **0.920** | **0.958** | **0.856** | 24 | **0.768** | **0.802** |
| | 336 | **0.812** | **0.920** | 0.776 | **0.882** | **0.912** | **0.926** | 0.839 | 48 | 0.765 | **0.775** |
| | 720 | **0.781** | **0.896** | 0.766 | **0.886** | **0.890** | **0.844** | 0.820 | 96 | **0.815** | **0.777** |
| | Avg | **0.820** | **0.925** | **0.793** | **0.897** | **0.913** | **0.927** | **0.833** | Avg | **0.785** | **0.798** |
| S-Mamba | 96 | 0.853 | 0.947 | 0.825 | 0.909 | **0.930** | 0.970 | 0.814 | 12 | 0.792 | 0.836 |
| | 192 | 0.825 | 0.932 | 0.796 | **0.898** | **0.920** | 0.946 | 0.85 | 24 | 0.767 | 0.796 |
| | 336 | 0.808 | 0.916 | 0.768 | 0.874 | 0.910 | 0.915 | **0.841** | 48 | **0.768** | 0.768 |
| | 720 | 0.755 | 0.895 | 0.756 | 0.867 | 0.888 | 0.827 | 0.827 | 96 | 0.813 | 0.774 |
| | Avg | 0.810 | 0.922 | 0.786 | **0.887** | 0.912 | 0.914 | **0.833** | Avg | **0.785** | 0.793 |
| iTransformer | 96 | 0.851 | 0.947 | 0.826 | 0.909 | 0.925 | 0.970 | 0.816 | 12 | 0.785 | 0.829 |
| | 192 | 0.827 | 0.930 | 0.799 | 0.877 | 0.918 | 0.946 | 0.851 | 24 | 0.748 | 0.780 |
| | 336 | 0.806 | 0.915 | 0.769 | 0.875 | 0.910 | 0.916 | 0.840 | 48 | 0.733 | 0.725 |
| | 720 | **0.781** | 0.892 | 0.755 | 0.869 | 0.887 | 0.826 | 0.821 | 96 | 0.787 | 0.696 |
| | Avg | 0.816 | 0.921 | 0.787 | 0.855 | 0.910 | 0.914 | 0.832 | Avg | 0.763 | 0.757 |

### 6.7 EFFICIENCY (Q6)

This section evaluates the computational efficiency of our proposed framework, FLDmamba, in comparison to several state-of-the-art baselines, including AutoFormer, RLinear, iTransformer, and S-Mamba. We assess efficiency on the ETTh1 and ETTh2 datasets, considering both training time per epoch and GPU memory consumption. The results, presented in Figure 12, demonstrate the following: **Comparative Efficiency of FLDmamba:** Our method, FLDmamba, exhibits a favorable balance between performance and computational efficiency, achieving comparable training times

and GPU memory costs to baselines. **Efficiency of Mamba-Based Methods:** Mamba-based methods, including FLDmamba and S-Mamba, demonstrate a compelling advantage in terms of training time and GPU memory consumption compared to Transformer-based baselines such as AutoFormer. This suggests that Mamba-based architectures offer a more efficient approach for handling time series data. These findings highlight the computational efficiency of our proposed framework, FLDmamba, while also emphasizing the potential benefits of Mamba-based architectures for addressing computational resource constraints in time series modeling.

### 6.7.1 HYPERPARAMETER STUDY (Q7)

In this section, we aim to conduct a parameter study to evaluate the impact of important parameters on the performance of our model, FLDmamba. The results are presented in Figure 10. Specifically, we vary the number of FLDmamba layers within the range of $\{1, 2, 3, 4, 5\}$, the hidden size from $\{128, 256, 512, 1024, 2048\}$, and the learning rate from $\{5 \times 10^{-4}, 5 \times 10^{-5}, 5 \times 10^{-6}, 5 \times 10^{-7}, 5 \times 10^{-8}\}$. Based on the results, we provide a summary of observations regarding these three parameters and their effects on performance, measured by MSE and MAE metrics, as follows: **(1)** We examine the impact of FLDmamba layers on the performance of FLDmamba. We observe that FLDmamba achieves the best performance when the number of layers is set to 2. However, as we increase the number of FLDmamba layers, the performance starts to diminish. This suggests that additional layers may introduce an over-smoothing effect, which negatively affects the performance of FLDmamba. **(2)** We also conducted experiments to investigate the effect of hidden sizes on FLDmamba performance. We find that our model FLDmamba achieves the highest performance when the hidden size is set to 1024. This indicates that smaller hidden sizes may not provide sufficient information, while larger hidden sizes may introduce redundant information that hampers the performance of FLDmamba. **(3)** Furthermore, we examine the impact of the learning rate on performance and observe that our method FLDmamba achieves the best performance when the learning rate is set to $5 \times 10^{-6}$. Smaller or larger learning rates may result in insufficient convergence or overfitting, which adversely affects the performance.

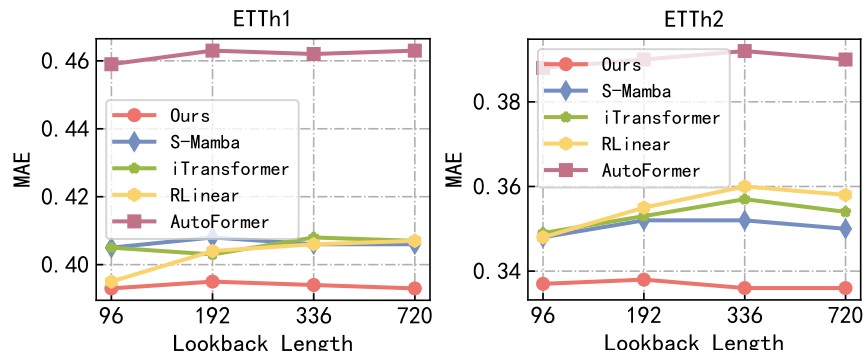

Figure 11: Long-term prediction with the lookback length from the range [96, 192, 336, 720].

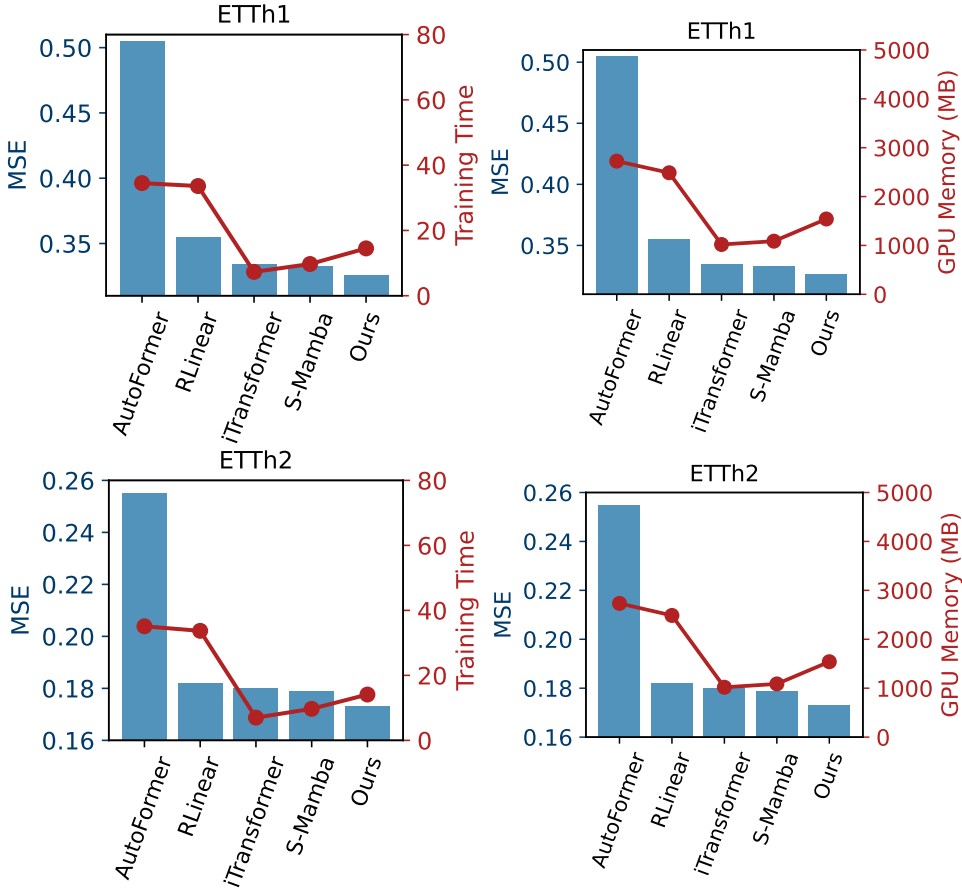

Figure 12: Model efficiency comparison on ETTh1 and ETTh2. The batch size is 32.

### 6.8 LIMITATIONS AND FUTURE WORK

The limitation of our work involves potential challenges in scaling the proposed model to extremely large datasets. Future efforts will focus on improving the model's adaptability to dynamic data environments and assessing its performance across diverse time series datasets. Furthermore, the exploration of alternative kernel functions beyond the RBF and a thorough scalability analysis will be pursued. Lastly, extending the model to accommodate missing data and integrating uncertainty quantification in predictions will bolster its practical utility.

### 6.9 LONG LOOKBACK COMPARISON

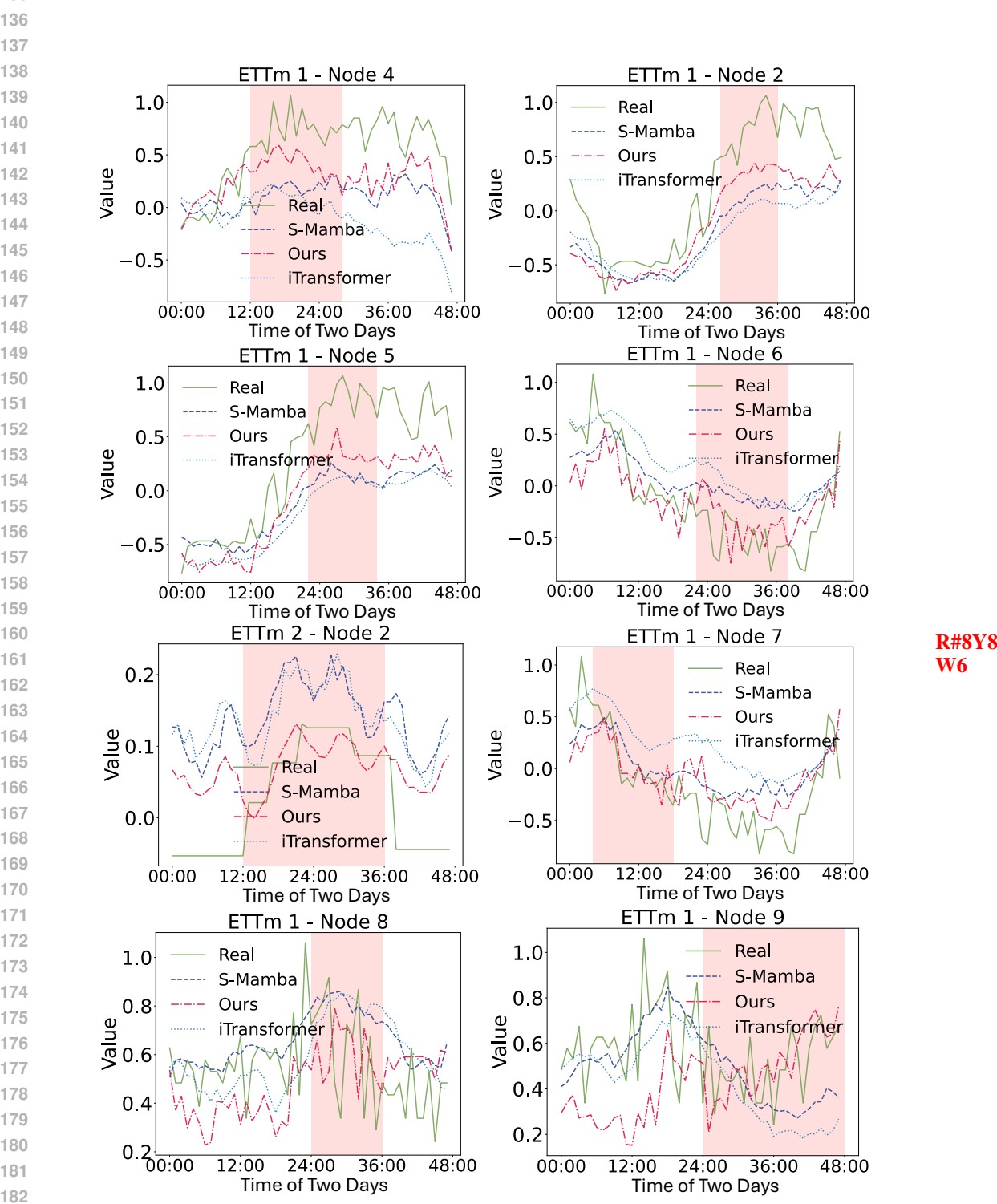

Figure 13: Case study of FLDmamba in terms of multi-scale periodicity.

To evaluate the performance of various models on long lookback, we conducted experiments using a lookback length of 1500 on the ETTh1 and ETTh2 datasets. Table 4 shows the MSE and MAE metrics for our proposed FLDmamba method, as well as other baseline models like S-Mamba, iTransformer, Rlinear, and AutoFormer. The results demonstrate that our FLDmamba method outperforms the other baselines across both datasets, highlighting its superior predictive capabilities.

Table 4: Performance of comparison when lookback length is set as 1500.

R#RZwJ-W2

|  | ETTh1 | | ETTh2 | |
|---|---|---|---|---|
|  | MSE | MAE | MSE | MAE |
| FLDmamba (ours) | **0.664** | **0.570** | **0.517** | **0.504** |
| S-Mamba | 0.715 | 0.603 | 0.539 | 0.522 |
| iTransformer | 0.787 | 0.634 | 0.549 | 0.528 |
| Rlinear | 1.281 | 0.884 | 3.015 | 1.366 |
| AutoFormer | 0.687 | 0.614 | 0.648 | 0.575 |

## 6.10 IMPACT OF RBF AND ILT

Table 5 presents comprehensive results of the Autoformer, Autoformer+RBF, and Autoformer+ILT models on the ETTh1 and ETTh2 datasets. The lookback length is fixed at 96, and the forecast length $T$ varies across 96, 192, 336, and 720. The bold font denotes the best model, and the underline denotes the second-best model. The results demonstrate that combination RBF and ILT with AutoFormer does not have positive impact on performance. This can be attributed to the redundant attention mechanism, which fails to demonstrate its advantages in the frequency domain.

R#Xe6T-W1, R#vEmK-W1,R#vEmK-W3

Table 5: We present comprehensive results of Autoformer, Autoformer+RBF, and Autoformer+ILT on the ETTh1 and ETTh2 datasets. The lookback length $L$ is fixed at 96, and the forecast length $T$ varies across 96, 192, 336, and 720. Bold font denotes the best model and underline denotes the second best.

| Models | | Autoformer | | Autoformer+RBF | | Autoformer+ILT | |
|---|---|---|---|---|---|---|---|
| Metric | | MSE | MAE | MSE | MAE | MSE | MAE |
| ETTh1 | 96 | 0.449 | 0.459 | **0.427** | **0.443** | 0.457 | 0.469 |
| | 192 | **0.500** | **0.482** | 0.501 | 0.484 | 0.522 | 0.503 |
| | 336 | **0.521** | **0.496** | 0.548 | 0.509 | 0.559 | 0.546 |
| | 720 | **0.514** | **0.512** | 0.537 | 0.526 | 0.543 | 0.534 |
| | Avg | **0.496** | **0.487** | 0.503 | 0.490 | 0.520 | 0.513 |
| ETTh2 | 96 | **0.358** | **0.397** | 0.360 | 0.401 | 0.454 | 0.473 |
| | 192 | **0.429** | **0.439** | 0.429 | 0.439 | 0.577 | 0.543 |
| | 336 | 0.496 | 0.487 | **0.467** | **0.474** | 0.668 | 0.596 |
| | 720 | **0.463** | **0.474** | 0.465 | 0.479 | 0.902 | 0.693 |
| | Avg | 0.437 | 0.449 | **0.430** | **0.448** | 0.650 | 0.576 |

## 6.11 COMPUTATIONAL OVERHEAD COMPARISON

Table 8 compares the time and memory consumption of different models on Electricity dataset. Specifically, it shows the runtime in seconds and the required RAM in MiB for Mamba+FFT, Mamba+ILT, our proposed method, S-Mamba, iTransformer, Autoformer, and Rlinear. The results demonstrate the computational efficiency of the proposed method, which achieves a good balance between inference time and memory usage compared to the other models.

## 6.12 OTHER KERNEL EXPERIMENTS

Table 6: Comprehensive results of PatchTST, PatchTST+RBF, and PatchTST+ILT on the ETTh1 and ETTh2 datasets. The lookback length $L$ is fixed at 96, and the forecast length $T$ varies across 96, 192, 336, and 720. Bold font denotes the best model and underline denotes the second best. **R#Xe6T-Q1**

| Models | | PatchTST | | PatchTST+RBF | | PatchTST+ILT | |
|---|---|---|---|---|---|---|---|
| Metric | | MSE | MAE | MSE | MAE | MSE | MAE |
| ETTh1 | 96 | 0.414 | 0.419 | 0.780 | 0.677 | 0.399 | 0.428 |
| | 192 | 0.460 | 0.445 | 0.913 | 0.743 | 0.465 | 0.461 |
| | 336 | 0.501 | 0.446 | 0.860 | 0.711 | 0.510 | 0.480 |
| | 720 | 0.500 | 0.488 | 0.883 | 0.726 | 0.568 | 0.535 |
| | Avg | 0.469 | 0.450 | 0.859 | 0.714 | 0.485 | 0.476 |
| ETTh2 | 96 | 0.302 | 0.348 | 1.338 | 0.874 | 0.359 | 0.394 |
| | 192 | 0.388 | 0.400 | 1.383 | 0.883 | 0.486 | 0.526 |
| | 336 | 0.426 | 0.433 | 1.415 | 0.892 | 0.538 | 0.499 |
| | 720 | 0.431 | 0.446 | 1.401 | 0.890 | 0.912 | 0.673 |
| | Avg | 0.387 | 0.407 | 1.384 | 0.885 | 0.574 | 0.523 |

Table 7: Comprehensive results of RLinear, RLinear+RBF, and RLinear+ILT on the ETTh1 and ETTh2 datasets. The lookback length $L$ is fixed at 96, and the forecast length $T$ varies across 96, 192, 336, and 720. Bold font denotes the best model and underline denotes the second best. **R#Xe6T-Q1**

| Models | | RLinear | | RLinear+RBF | | RLinear+ILT | |
|---|---|---|---|---|---|---|---|
| Metric | | MSE | MAE | MSE | MAE | MSE | MAE |
| ETTh1 | 96 | 0.386 | 0.395 | 0.501 | 0.469 | 0.384 | 0.402 |
| | 192 | 0.437 | 0.424 | 0.537 | 0.490 | 0.429 | 0.426 |
| | 336 | 0.479 | 0.446 | 0.567 | 0.507 | 0.462 | 0.445 |
| | 720 | 0.481 | 0.470 | 0.565 | 0.528 | 0.463 | 0.463 |
| | Avg | 0.446 | 0.434 | 0.543 | 0.499 | 0.435 | 0.434 |
| ETTh2 | 96 | 0.288 | 0.338 | 0.359 | 0.393 | 0.307 | 0.355 |
| | 192 | 0.374 | 0.390 | 0.434 | 0.435 | 0.387 | 0.402 |
| | 336 | 0.415 | 0.461 | 0.462 | 0.460 | 0.424 | 0.434 |
| | 720 | 0.420 | 0.440 | 0.459 | 0.466 | 0.424 | 0.443 |
| | Avg | 0.374 | 0.407 | 0.428 | 0.438 | 0.385 | 0.409 |

From the results presented in Table 9, we observe that the RBF (Radial Basis Function) kernel achieves the best performance on time series prediction compared to the Laplacian and Sigmoid kernels. This can be attributed to the inherent ability of the RBF kernel to capture the nonlinear and complex patterns in the time series data more effectively.

### 6.13 VISUALIZATION

We show visualization of $\Delta A$ and $\Delta_F A$ as follows. This figure visualizes the differences between $\Delta A$ and $\Delta_F A$ over time on ETTm1. $\Delta A$ represents the change in absorbance, while $\Delta_F A$ represents the change in fluorescence absorbance. The figure shows the fluctuations in these two measures, highlighting their distinct patterns over the duration of the experiment. **R#vEmK-W4**

### 6.14 ADDITIONAL TABLE OF ABLATION STUDY

Table 8: Comparison of different models in terms of time and memory consumption on Electricity.

|  | Mamba+FFT | Mamba+ILT | Ours | S-Mamba | iTransformer | AutoFormer | Rlinear |
|---|---|---|---|---|---|---|---|
| Time (Seconds) | 2.565e-3 | 2.274e-3 | 2.984e-3 | 2.999e-3 | 1.869e-3 | 8.975e-3 | 5.345e-3 |
| RAM (MiB) | 564 | 562 | 568 | 566 | 560 | 596 | 588 |

Table 9: Performance comparison of different kernels with MSE and MAE.

|  | RBF | | Laplacian | | Sigmoid | |
|---|---|---|---|---|---|---|
|  | MSE | MAE | MSE | MAE | MSE | MAE |
| 96 | 0.374 | 0.393 | 0.383 | 0.402 | 0.384 | 0.402 |
| 192 | 0.427 | 0.422 | 0.446 | 0.434 | 0.445 | 0.434 |
| 336 | 0.447 | 0.441 | 0.488 | 0.460 | 0.486 | 0.459 |
| 720 | 0.469 | 0.463 | 0.504 | 0.484 | 0.502 | 0.483 |
| Avg | 0.434 | 0.430 | 0.45525 | 0.445 | 0.454 | 0.445 |

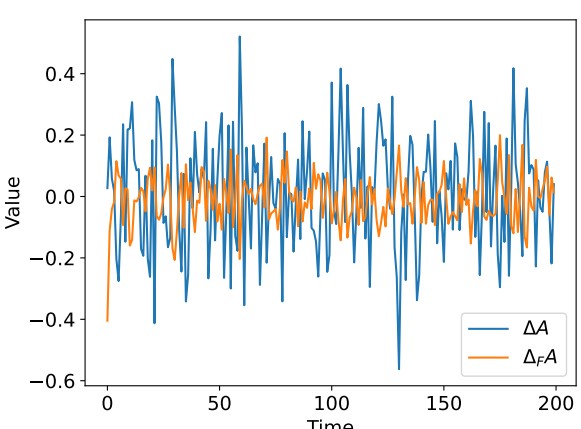

Figure 14: Visualization of $\Delta A$ and $\Delta_F A$

Table 10: Ablation study PeMS08 and Exchange datasets.

| PeMS08 | -FT | -FM | -Ma | -RBF | -ILT | Ours |
|---|---|---|---|---|---|---|
| MSE | 0.291 | 0.306 | 0.353 | 0.277 | 0.314 | 0.243 |
| MAE | 0.341 | 0.351 | 0.382 | 0.332 | 0.358 | 0.305 |
| Exchange | -FT | -FM | -Ma | -RBF | -ILT | Ours |
| MSE | 0.090 | 0.090 | 0.089 | 0.092 | 0.098 | 0.085 |
| MAE | 0.216 | 0.217 | 0.214 | 0.219 | 0.223 | 0.205 |

