# OpenReview forum: "FLDmamba:  Integrating Fourier and Laplace Transform Decomposition with Mamba for Enhanced Time Series Prediction"
_ICLR.cc/2025/Conference — Submitted to ICLR 2025_

### Official Review · Reviewer_6yv6 · 2024-10-30

**Soundness:** 3
**Presentation:** 3
**Contribution:** 3
**Rating:** 6
**Confidence:** 3

**Summary:**

The paper proposes FLDmamba, a novel framework that integrates Fourier and Laplace Transform Decomposition with the Mamba State-Space Model (SSM) to enhance long-term time series prediction. The authors identify key challenges in existing models, particularly in capturing multi-scale periodicity, transient dynamics, and handling data noise. Extensive experiments on nine real-world datasets demonstrate that FLDmamba outperforms state-of-the-art Transformer-based and Mamba-based architectures

**Strengths:**

1. The paper introduces a novel integration of Fourier and Laplace transforms into the Mamba framework, addressing the limitations of previous SSMs in capturing multi-scale periodicity and transient dynamics.
2. The paper includes thorough experiments on nine diverse real-world datasets, covering various domains. The results consistently show that FLDmamba achieves superior performance compared to strong baselines.
3. The model's robustness to data noise is evaluated, showing that FLDmamba maintains high performance even under increased noise levels, outperforming other methods like S-Mamba and iTransformer. Detailed ablation studies are conducted to isolate and demonstrate the contribution of each component in the FLDmamba framework.

**Weaknesses:**

1. While the paper explains the intuition behind using the Laplace transform to capture transient dynamics, it lacks a deeper theoretical exploration of how exactly the inverse Laplace transform contributes to performance improvements in the context of the model.
2. The experimental comparison focuses primarily on Transformer-based models and Mamba-based methods. Inclusion of more diverse SSM-based baselines, such as those based on S4 or other recent advances, would strengthen the evaluation.

**Questions:**

1. Can you provide more details on how the inverse Laplace transform is computed in practice within your framework? Given that inverse Laplace transforms can be numerically challenging, how do you ensure stability and efficiency in this component?
2. Have you explored using alternative kernel functions beyond the RBF kernel for data smoothing? If so, how do they compare in terms of performance and computational cost?

---

> ### Author Response · Authors · 2024-11-21
>
> >W1. While the paper explains the intuition behind using the Laplace transform to capture transient dynamics, it lacks a deeper theoretical exploration of how exactly the inverse Laplace transform contributes to performance improvements in the context of the model.
>
> >Q1. Can you provide more details on how the inverse Laplace transform is computed in practice within your framework? Given that inverse Laplace transforms can be numerically challenging, how do you ensure stability and efficiency in this component?
>
> Thanks for your comment. We provide theoretical explanation in Appendix 6.2, which we summarize here. Transient dynamics are characterized by exponential decaying amplitudes w.r.t. time $t$. Thus, a time series variable $u(t)$ exhibiting transient dynamics can then in general be decomposed by
>
> $u(t)=\sum_{n=1}^M A_n e^{-\xi_n t}\cos(\omega_n t + \varphi_n)$  (Eq. 20)
>
> where $\xi_n, n=1,2,...$ are the decaying rates, $\omega_n, n=1,2,...$ are the corresponding periodic frequencies (can be 0 for non-periodic signal), $A_n, \varphi_n, n=1,2,...$ are the amplitudes and phases, respectively.
>
> When we are performing prediction on time series, we are essentially learning an operator (mapping between functions) that maps a segment of time series $u(t), t\in[t_0,t_1]$ in the past, to a segment of time series $u(t), t\in[t_1,t_2]$ in the future. Thus, the above $A_n$, $\lambda_n$, and $\omega_n$ are in general functions of the past time series $u(t), t\in[t_0,t_1]$. Below, we show how our modeling of inverse Laplace transform exactly captures the above transient dynamics (Eq. 20).
>
> As was explained in Appendix 6.2 in the original submission, we model the operator which maps an input function $v(t)$ to an output function $u(t)$ as
>
> $u(t)=(\kappa(\phi)*v)(t)=\int_D \kappa_\phi(t-\tau)v(\tau)d\tau$
>
> Performing Laplace transform on both sides, we have
>
> $U(s)=K_\phi(s)V(s)$
>
> where $K_\phi(s)=\mathcal{L}\{\kappa_\phi(t)\}$ and $V(s)=\mathcal{L}\{v(t)\}$, $U(s)=\mathcal{L}\{u(t)\}$. Based on the Residue Theorem in complex analysis, the poles (singularities) in the complex plane determines its behavior in the original space. Therefore, we assume $K_\phi(s)=\sum_{n=1}^N \frac{\beta_n}{s-\mu_n}$ in the Laplace space, where $\beta_n\in \mathbb{R}$ and $\mu_n\in \mathbb{C}$ are learnable parameters. Also, performing Fourier series expansion on $v(t)$, we have $v(t)=\sum_{l=-\infty}^{\infty}\alpha_l \exp{i \omega_l t}$, which results in $V(s)=\sum_{l=-\infty}^{\infty}\frac{\alpha_l}{s-i\omega_l}$. Mapping back into the original space, we have
>
>
> $u(t)=\sum_{n=1}^N\gamma_n \exp(\mu_n t)+ \sum_{l=-\infty}^{\infty}\lambda_l \exp(i\omega_l t)$        (Eq. 19 in Appendix 6.2)
>
> Here $\gamma_n$, $\lambda_l$ are derived parameters from $\beta_n$, $\mu_n$, $\omega_l$ and $\alpha_l$, the former two depending on the kernel $\kappa(\phi)$, and the latter two depending on $v(t)$.
>
> Note that $\mu_n\in\mathbb{C}$ is a complex number, whose real part and imagery part represent decaying and periodic behaviors, respectively. If we truncate the number of Fourier series terms $l$, the above Eq. (19) reduces to
>
> $u(t)=\sum_{n=1}^M A_n e^{-\sigma_n t}\cos(w_n t + \varphi_n)$ (Eq. 21)
>
> In our work, we directly parameterize the above $A_n$, $\sigma_n$, $w_n$, and $\varphi_n$ as learnable functions of the output of the previous layer, which in turn are functions of the history time series. We see that this equation (Eq. 21) exactly matches the above Eq. 20 which **characterizes transient dynamics**. Therefore, our parameterization of the inverse Laplace transform via Eq. 21 can learn transient dynamics accurately. Furthermore, in contrast to performing inverse Laplace transform which involves integration in the complex plane where the integrand has poles, we see that our parameterization in Eq. 21 has better efficiency and stability.
>
> We have updated Appendix 6.2 to include the above analysis.

---

> ### Author Response · Authors · 2024-11-21
>
> >W2. The experimental comparison focuses primarily on Transformer-based models and Mamba-based methods. Inclusion of more diverse SSM-based baselines, such as those based on S4 or other recent advances, would strengthen the evaluation.
>
> Thanks for your point. We have conducted new Mamba-based methods including SST and Bi-Mamba+. Results are shown in the following table:
>
> | Models       | Metric | FLDmamba (MSE) | FLDmamba (MAE) | SST (MSE) | SST (MAE) | Bi-Mamba+ (MSE) | Bi-Mamba+ (MAE) |
> |--------------|--------|----------------|----------------|-----------|-----------|-----------------|-----------------|
> | **ETTm1**    | 96     | **0.318**      | **0.360**      | 0.337     | 0.374     | 0.355           | 0.386           |
> |              | 192    | **0.365**      | **0.384**      | 0.377     | 0.392     | 0.415           | 0.419           |
> |              | 336    | 0.404          | **0.409**      | 0.401     | 0.412     | 0.450           | 0.442           |
> |              | 720    | _0.464_        | _0.441_        | 0.498     | 0.464     | 0.497           | 0.476           |
> |              | Avg    | _0.389_        | **0.399**      | 0.413     | 0.411     | 0.429           | 0.431           |
> | **ETTm2**    | 96     | **0.173**      | **0.253**      | 0.185     | 0.274     | 0.186           | 0.278           |
> |              | 192    | **0.240**      | **0.299**      | 0.248     | 0.313     | 0.257           | 0.324           |
> |              | 336    | **0.301**      | **0.307**      | 0.309     | 0.351     | 0.318           | 0.362           |
> |              | 720    | **0.401**      | **0.397**      | 0.406     | 0.405     | 0.412           | 0.416           |
> |              | Avg    | **0.279**      | **0.314**      | 0.287     | 0.333     | 0.293           | 0.347           |
> | **ETTh1**    | 96     | **0.374**      | **0.393**      | 0.390     | 0.403     | 0.398           | 0.416           |
> |              | 192    | _0.427_        | **0.422**      | 0.451     | 0.438     | 0.451           | 0.446           |
> |              | 336    | **0.447**      | **0.441**      | 0.496     | 0.458     | 0.497           | 0.473           |
> |              | 720    | **0.469**      | **0.463**      | 0.520     | 0.493     | 0.526           | 0.509           |
> |              | Avg    | **0.434**      | **0.430**      | 0.439     | 0.448     | 0.468           | 0.461           |
> | **ETTh2**    | 96     | **0.287**      | **0.337**      | 0.298     | 0.351     | 0.307           | 0.363           |
> |              | 192    | **0.370**      | **0.388**      | 0.393     | 0.407     | 0.394           | 0.414           |
> |              | 336    | **0.412**      | **0.425**      | 0.436     | 0.441     | 0.437           | 0.447           |
> |              | 720    | **0.419**      | **0.438**      | 0.431     | 0.449     | 0.445           | 0.462           |
> |              | Avg    | **0.372**      | **0.396**      | 0.390     | 0.412     | 0.396           | 0.422           |
> | **Electricity** | 96  | **0.137**      | **0.234**      | 0.192     | 0.280     | 0.146           | 0.246           |
> |              | 192    | **0.158**      | **0.251**      | 0.191     | 0.280     | 0.167           | 0.265           |
> |              | 336    | 0.182          | **0.173**      | 0.211     | 0.299     | 0.182           | 0.281           |
> |              | 720    | **0.200**      | **0.292**      | 0.264     | 0.340     | 0.208           | 0.304           |
> |              | Avg    | **0.170**      | **0.238**      | 0.215     | 0.300     | 0.176           | 0.274           |
> | **Exchange** | 96     | **0.085**      | **0.205**      | 0.091     | 0.216     | 0.103           | 0.233           |
> |              | 192    | **0.175**      | **0.297**      | 0.189     | 0.313     | 0.214           | 0.337           |
> |              | 336    | 0.317          | _0.407_        | 0.333     | 0.421     | 0.366           | 0.445           |
> |              | 720    | **0.825**      | **0.683**      | 0.916     | 0.729     | 0.931           | 0.738           |
> |              | Avg    | **0.351**      | **0.400**      | 0.382     | 0.420     | 0.404           | 0.428           |

---

> ### Author Response · Authors · 2024-11-21
>
> **(continued)**
>
> | Models       | Metric | FLDmamba (MSE) | FLDmamba (MAE) | SST (MSE) | SST (MAE) | Bi-Mamba+ (MSE) | Bi-Mamba+ (MAE) |
> |--------------|--------|----------------|----------------|-----------|-----------|-----------------|-----------------|
> | **Solar-Energy** | 96 | **0.202**      | **0.233**      | 0.238     | 0.277     | 0.231           | 0.286           |
> |              | 192    | **0.230**      | **0.254**      | 0.299     | 0.319     | 0.257           | 0.285           |
> |              | 336    | _0.254_        | **0.265**      | 0.310     | 0.327     | 0.256           | 0.293           |
> |              | 720    | _0.252_        | **0.271**      | 0.310     | 0.330     | 0.252           | 0.295           |
> |              | Avg    | _0.235_        | **0.256**      | 0.289     | 0.313     | 0.249           | 0.290           |
> | **PEMS04**   | 12     | **0.075**      | _0.182_        | 0.110     | 0.226     | 0.082           | 0.193           |
> |              | 24     | **0.084**      | **0.193**      | 0.161     | 0.275     | 0.099           | 0.214           |
> |              | 48     | **0.105**      | **0.217**      | 0.345     | 0.403     | 0.123           | 0.240           |
> |              | 96     | **0.130**      | **0.243**      | 0.588     | 0.553     | 0.151           | 0.267           |
> |              | Avg    | **0.099**      | **0.209**      | 0.301     | 0.364     | 0.114           | 0.229           |
> | **PEMS08**   | 12     | **0.075**      | **0.177**      | 0.099     | 0.214     | 0.080           | 0.190           |
> |              | 24     | **0.102**      | **0.207**      | 0.169     | 0.277     | 0.114           | 0.223           |
> |              | 48     | **0.154**      | **0.226**      | 0.274     | 0.360     | 0.175           | 0.271           |
> |              | 96     | _0.243_        | 0.305          | 0.522     | 0.499     | 0.298           | 0.348           |
> |              | Avg    | **0.145**      | 0.228          | 0.266     | 0.338     | 0.167           | 0.258           |
>
>
> The results indicate that our method outperforms other Mamba-based baselines. This improvement is attributed to the incorporation of FFT and ILT, which effectively capture multi-scale periodicity and transient dynamics.
> **Results are also added in Table 1 in revised version.**  Please refer to the revised version.

---

> > ### Author Response · Authors · 2024-11-21
> >
> > >Q2. Have you explored using alternative kernel functions beyond the RBF kernel for data smoothing? If so, how do they compare in terms of performance and computational cost?
> >
> > Thanks for your comment. To address your concern, we have conducted  experiments via replacing the RBF kernel with Laplacian and Sigmoid kernels. The results are shown in the following table:
> >
> >
> > || RBF(MSE)   | RBF(MAE)  | Laplacian(MSE)  | Laplacian(MAE)     | Sigmoid(MSE)     | Sigmoid(MAE)    |
> > |:-----:|:-------:|:-----------:|:---------:|:---------:|:---------:|:----:|
> > | 96  | 0.374 | 0.393     | 0.383   | 0.402   | 0.384   | 0.402  |
> > | 192 | 0.427 | 0.422     | 0.446   | 0.434   | 0.445   | 0.434  |
> > | 336 | 0.447 | 0.441     | 0.488   | 0.46    | 0.486   | 0.459  |
> > | 720 | 0.469 | 0.463     | 0.504   | 0.484   | 0.502   | 0.483  |
> > | Avg | 0.434 | 0.43      | 0.45525 | 0.445   | 0.45425 | 0.4445 |
> >
> >
> > From above results, we see that RBF kernel achieves the best performance on time series prediction than Laplacian and Sigmoid kernels. This can be attributed to the inherent ability of the RBF kernel to capture the nonlinear and complex patterns in the time series data more effectively.

---

> > > ### Author Response · Authors · 2024-11-25
> > >
> > > Dear Reviewer,
> > >
> > > We believe that the additional information we provided in our rebuttal—such as new experimental results, further details, and clarifications on misunderstandings—addresses your key questions. Please let us know if our response has adequately addressed your concerns. We are more than willing to discuss any points that may still be unclear. We hope that the improvements and clarifications provided in our response will positively influence your assessment of our work.
> > >
> > > Best, Authors of Paper 4022

---

### Official Review · Reviewer_8Y8C · 2024-10-30

**Soundness:** 4
**Presentation:** 3
**Contribution:** 4
**Rating:** 8
**Confidence:** 3

**Summary:**

The paper presents FLDmamba, a novel time series prediction framework combining Fourier and Laplace transformations with the Mamba architecture to improve accuracy and robustness for long-term predictions. The Fourier transform aims to capture multi-scale periodicity and reduce noise, while the Laplace transform enhances the model’s ability to capture transient dynamics. Through ablation studies and benchmark comparisons, FLDmamba demonstrates state-of-the-art performance on several time series prediction datasets. The authors also evaluate the model's robustness, efficiency, and sensitivity to hyperparameters, contributing to a well-rounded evaluation.

**Strengths:**

1. Innovative Framework: Integrating Fourier and Laplace transformations into the Mamba model is novel in the context of time series forecasting. This combination allows FLDmamba to address core challenges in time series data—multi-scale periodicity, noise reduction, and transient dynamics.

2. Solid Performance Gains: FLDmamba consistently outperforms other models on key benchmarks, particularly in scenarios involving noisy data or long lookback lengths, demonstrating that the model effectively generalizes across diverse datasets.

**Weaknesses:**

While this paper is generally strong, there are a few minor weaknesses that could be addressed to further strengthen the contribution:

1. Incomplete Justification for RBF Kernel: Although the RBF kernel is presented as an effective data-smoothing technique, its choice is not empirically validated. A comparison with other kernel functions or a focused ablation study would help verify this choice and ensure that RBF is the optimal choice.

2. Unclear Necessity of FFT-IFFT Sequence: The FMamba block employs an FFT followed by an IFFT without a clear explanation of any specific frequency-domain manipulations before reconstructing the signal in the time domain. If this process is meant to filter specific frequencies or reduce noise, the details of such operations should be specified. Otherwise, the sequence could appear redundant, as it may be feasible for the neural network to approximate frequency characteristics without explicitly embedding FFT.

3. Limited Explanation of Explicit Transformations: While the inclusion of Fourier and Laplace transforms is well-motivated theoretically, it remains unclear why these explicit transformations are necessary. Neural networks, particularly those with linear layers, can approximate operations like FFT. A clearer discussion on the unique advantages of explicitly integrating these transforms would strengthen the architectural justification.

4. Incomplete Complexity Analysis: The complexity analysis, which estimates FLDmamba’s time complexity as $𝑂(𝐵𝐿𝑉𝑁)$, does not fully account for the computational costs of FFT, IFFT, and inverse Laplace transforms. Each of these operations introduces additional costs (e.g., $O(BLNlogL)$ for FFT)) that may not scale efficiently for large datasets. This makes the current complexity analysis potentially optimistic, particularly given that working in the complex domain could introduce additional memory and processing overhead. Wall-clock inference times compared to baseline models would better validate FLDmamba's practical efficiency and help justify the complexity of the FFT and Laplace operations.

5. Incomplete Citation of S-Mamba: While S-Mamba is frequently referenced as a baseline, it lacks a formal citation in the main text. Adding this citation would improve the academic rigor and proper attribution within the paper.

6. Clarity of Figures: Some figures could benefit from clearer axis labels to improve interpretability. For example, in Figure 1, the y-axis label is ambiguous, and the x-axis label as “Time of Day” is potentially misleading since it exceeds 24 hours. Clarifying these points would improve the readability of the time series prediction results.

**Questions:**

1. Since both FFT and Discrete Cosine Transform (DCT) are effective for frequency-domain analysis, could the authors clarify why they selected FFT over DCT? DCT, for instance, has shown advantages in signal compression and noise reduction and might benefit time series forecasting by emphasizing low-frequency components. Further insight on this choice would help clarify the design decision.

3. Deep learning models with linear layers can often approximate linear transformations, including FFT. Could the authors elaborate on the specific necessity of explicitly embedding Fourier and Laplace transforms rather than relying on the model's intrinsic capacity to learn these linear relationships? This would clarify whether these transformations improve interpretability, robustness, or training efficiency in ways that the network alone might not achieve.

---

> ### Author Response · Authors · 2024-11-21
>
> >W1. Incomplete Justification for RBF Kernel: Although the RBF kernel is presented as an effective data-smoothing technique, its choice is not empirically validated. A comparison with other kernel functions or a focused ablation study would help verify this choice and ensure that RBF is the optimal choice.
>
>
>
> Thanks for your comment. To address your concern, we have conducted  experiments by replacing the RBF kernel with Laplacian and Sigmoid kernels. And the results are shown in the following table:
>
>
> || RBF(MSE)   | RBF(MAE)  | Laplacian(MSE)  | Laplacian(MAE)     | Sigmoid(MSE)     | Sigmoid(MAE)    |
> |:-----:|:-------:|:-----------:|:---------:|:---------:|:---------:|:----:|
> | 96  | 0.374 | 0.393     | 0.383   | 0.402   | 0.384   | 0.402  |
> | 192 | 0.427 | 0.422     | 0.446   | 0.434   | 0.445   | 0.434  |
> | 336 | 0.447 | 0.441     | 0.488   | 0.46    | 0.486   | 0.459  |
> | 720 | 0.469 | 0.463     | 0.504   | 0.484   | 0.502   | 0.483  |
> | Avg | 0.434 | 0.43      | 0.45525 | 0.445   | 0.45425 | 0.4445 |
>
>
> From above results, we observe that RBF kernel achieves the best performance on time series prediction than Laplacian and Sigmoid kernels. This can be attributed to the inherent ability of the RBF kernel to capture the nonlinear and complex patterns in the time series data more effectively.
>
>
>
> >W2. Unclear Necessity of FFT-IFFT Sequence: The FMamba block employs an FFT followed by an IFFT without a clear explanation of any specific frequency-domain manipulations before reconstructing the signal in the time domain. If this process is meant to filter specific frequencies or reduce noise, the details of such operations should be specified. Otherwise, the sequence could appear redundant, as it may be feasible for the neural network to approximate frequency characteristics without explicitly embedding FFT.
>
> Thanks for your comment. To address the concern regarding the necessity of the FFT-IFFT sequence in the FMamba block, it is important to clarify the intended purpose of this process. The sequence of applying the FFT followed by the IFFT is designed to facilitate specific frequency-domain manipulations that are crucial for enhancing the model's performance. (1) The FFT allows us to analyze the frequency components of an input signal by transforming it from the time domain to the frequency domain using $\Delta' = \text{FFT}(\Delta)$. We then apply $\Delta_F = \text{IFFT}(\tilde{W} \cdot \Delta')$ to return to the time domain. Here $\tilde{W}$ is the Fourier transform of the kernel $\tilde{K}$. In this paper, we treat $\tilde{W}$ as a learnable parameter matrix. This process helps us identify and filter out specific frequencies that may introduce noise, ultimately enhancing the signal and improving forecasting accuracy [1].
> (2) The FFT-IFFT sequence can be effectively employed to isolate and mitigate unwanted frequency components. This process ensures that the reconstructed signal retains the essential features while minimizing the influence of noise, leading to more robust predictions.
> (3) While neural networks can learn to approximate frequency characteristics, explicitly embedding the FFT provides a structured approach to feature extraction. This allows for a more interpretable representation of the data, highlighting important frequency components that may not be as easily captured through learning alone.
> (4) The FFT-IFFT sequence serves as a complementary mechanism that enhances the neural network's ability to learn complex temporal patterns. By combining explicit frequency analysis with the neural network's learning capabilities, we create a more powerful model that effectively captures both global and local dynamics in the data.
>
> The FFT-IFFT sequence is a deliberate choice aimed at enabling specific frequency-domain manipulations, such as filtering and noise reduction, which ultimately enhance the model's performance. We will clarify these points in the revised version to ensure a better understanding of its necessity.
>
> [1] Li, Zongyi, et al. "Fourier neural operator for parametric partial differential equations."

---

> > ### Author Response · Authors · 2024-11-21
> >
> > >W3. Limited Explanation of Explicit Transformations: While the inclusion of Fourier and Laplace transforms is well-motivated theoretically, it remains unclear why these explicit transformations are necessary. Neural networks, particularly those with linear layers, can approximate operations like FFT. A clearer discussion on the unique advantages of explicitly integrating these transforms would strengthen the architectural justification.
> >
> >
> > Thanks for your comment. (1) The Fourier and Laplace transforms offer domain-specific insights into the frequency and time-domain characteristics of the data, respectively. By explicitly incorporating these transforms, the model gains a more interpretable representation of the underlying patterns in the data, which can enhance model understanding and decision-making. (2) While neural networks can approximate certain operations, explicitly incorporating Fourier and Laplace transforms can enhance the efficiency of capturing frequency components and transient behaviors in the data. This explicit modeling can lead to more efficient learning and better generalization to unseen data patterns, especially in scenarios where these specific characteristics are crucial for accurate predictions.
> >
> > While neural networks can approximate certain operations like FFT, the explicit integration of Fourier and Laplace transforms in neural network architectures offers unique advantages in terms of interpretability, feature extraction, efficiency, complex pattern detection, and leveraging complementary capabilities. These benefits collectively contribute to a more robust and specialized modeling approach that can better capture the intricate characteristics of time series data, ultimately improving the model's performance and adaptability in handling diverse temporal patterns.
> >
> >
> > >W4. Incomplete Complexity Analysis: The complexity analysis, which estimates FLDmamba’s time complexity as
> > , does not fully account for the computational costs of FFT, IFFT, and inverse Laplace transforms. Each of these operations introduces additional costs (e.g.,
> >  for FFT)) that may not scale efficiently for large datasets. This makes the current complexity analysis potentially optimistic, particularly given that working in the complex domain could introduce additional memory and processing overhead. Wall-clock inference times compared to baseline models would better validate FLDmamba's practical efficiency and help justify the complexity of the FFT and Laplace operations.
> >
> >
> >
> > Thanks for your point. In the model complexity analysis, we have correctly estimate the complexity of FFT, inverse Laplace transform, etc. with the big O notation. The reviewer did raise a good point that there may be additional costs, leading to constant overhead or large coefficients which is not measured by the big O notation. To empirically verify, we provide experiments of inference time of ours, Mamba + FFT inference time, inference time of Mamba+ Laplace operation, Mamba + FFT + Laplace operation inference time, AutoFormer inference time, RLiniear inference time, iTransformer inference time in the following table:
> >
> >
> > |     | Mamba+FFT   | Mamba+ILT   | Ours        | S-Mamba     | iTransformer | Autoformer  | Rlinear     |
> > |:---------:|:-------------:|:-------------:|:------------:|:-------------:|:--------------:|:-------------:|:-------------:|
> > | Time/s  | 2.565e-3 | 2.274e-3 | 2.984e-3 | 2.999e-3 | 1.869e-3  | 8.975e-3 | 5.345e-3 |
> > | RAM/MiB | 564         | 562         | 568         | 566         | 566          | 596         | 588         |
> >
> >
> >
> >  We also added this part in section 6.11 in Appendix the revised version. Please refer to the updated manuscript.
> >
> >
> >
> >
> > >W5. Incomplete Citation of S-Mamba: While S-Mamba is frequently referenced as a baseline, it lacks a formal citation in the main text. Adding this citation would improve the academic rigor and proper attribution within the paper.
> >
> > Thanks for your comment. We have addded formal citation of S-Mamba in the revised version.
> >
> >
> > >W6. Clarity of Figures: Some figures could benefit from clearer axis labels to improve interpretability. For example, in Figure 1, the y-axis label is ambiguous, and the x-axis label as “Time of Day” is potentially misleading since it exceeds 24 hours. Clarifying these points would improve the readability of the time series prediction results.
> >
> > Thanks for your comment. We have revised figures in the revised version.

---

> > > ### Author Response · Authors · 2024-11-21
> > >
> > > >Q1.Since both FFT and Discrete Cosine Transform (DCT) are effective for frequency-domain analysis, could the authors clarify why they selected FFT over DCT? DCT, for instance, has shown advantages in signal compression and noise reduction and might benefit time series forecasting by emphasizing low-frequency components. Further insight on this choice would help clarify the design decision.
> > >
> > > Thanks for your comment. Firstly, FFT is commonly chosen for its ability to provide a detailed representation of both high and low-frequency components in the frequency domain. This comprehensive frequency analysis is crucial for capturing a wide range of patterns present in time series data, making FFT a versatile choice for modeling diverse temporal characteristics. Then, While DCT is known for its effectiveness in signal compression and noise reduction, FFT offers a more straightforward interpretation of frequency components in the data. The clearer separation of frequencies provided by FFT can aid in identifying periodic patterns and transient dynamics, which are essential for accurate time series forecasting.
> > >
> > >
> > > >Q2.Deep learning models with linear layers can often approximate linear transformations, including FFT. Could the authors elaborate on the specific necessity of explicitly embedding Fourier and Laplace transforms rather than relying on the model's intrinsic capacity to learn these linear relationships? This would clarify whether these transformations improve interpretability, robustness, or training efficiency in ways that the network alone might not achieve.
> > >
> > > Thanks for your comment. (1) Explicitly embedding Fourier and Laplace transforms can enhance the model's robustness by ensuring that essential domain-specific information is properly encoded in the model's representations. This explicit modeling approach can improve the model's ability to generalize to unseen data patterns and enhance its resilience to noise and variability in the input data. (2) While deep learning models can approximate linear transformations like FFT, explicitly integrating Fourier and Laplace transforms can streamline the learning process by providing a structured framework for capturing frequency and time-domain features. This structured approach can potentially reduce the computational complexity of the learning task and improve training efficiency by focusing the model's attention on relevant features, like FNO[1] in Neural PDE and LNO[2] which explicitly incorporates Laplace analysis.
> > >
> > > [1] Li, Zongyi, et al. "Fourier neural operator for parametric partial differential equations."
> > >
> > > [2] Cao, Qianying, Somdatta Goswami, and George Em Karniadakis. "Laplace neural operator for solving differential equations." Nature Machine Intelligence 6.6 (2024): 631-640.

---

> > > > ### Author Response · Authors · 2024-11-25
> > > >
> > > > Dear Reviewer,
> > > >
> > > > We believe that the additional information we provided in our rebuttal—such as new experimental results, further details, and clarifications on misunderstandings—addresses your key questions. Please let us know if our response has adequately addressed your concerns. We are more than willing to discuss any points that may still be unclear.
> > > >
> > > > Best, Authors of Paper 4022

---

### Official Review · Reviewer_RZwJ · 2024-10-30

**Soundness:** 2
**Presentation:** 3
**Contribution:** 2
**Rating:** 5
**Confidence:** 4

**Summary:**

This paper introduces FLDMamba, a multi-variate time series prediction model.
The model focuses on (1) multi-resolution on the periodicity of input sequence, (2) Transient dynamics of the time series and (3) noise filtering in time series data.
The authors construct the FMamba-Mamba (FMM) layer as the foundational unit to build the FLDMamba model.
The authors conduct extensive experiments to show the effectiveness of the proposed model, model's capability on long-range prediction, and noise robustness.

**Strengths:**

- 1. I think the motivation of the paper is reasonable. Using the RBF kernel does seem to be a fair approach.
- 2. I also think the use of the FFT makes sense especially when dealing with the lead-lag relationships between variates. The convolution operation is able to reveal such information in the discrete data points.
- 3. The experiments contain most of the state-of-the-art time series prediction models I can think of.

**Weaknesses:**

- 1. My biggest concern about this paper is their evaluation metric. I believe using R2 score or Pearson correlation is more suitable for the task. However, this paper only considers the MSE and MAE error, while the MSE and MAE seems to be lower than all other baselines, I still have some doubts on the models ability to capture informative time series patterns.
- 2. The long-term prediction part doesn't seem to be very informative. Beside the problem on MSE and MAE, the max look-back length is only set to 720, which most baselines are capable of handling. And the improvement is small in my opinion.

- I do consider the technical details of this paper is sound and informative, I would love to increase my ratings as long as the R2 score and Pearson correlation also reflects the effectiveness of their model.

**Questions:**

- 1. Are you able to report the R2 score or the Pearson correlation? I strongly believe this is an essential metric the author should provide when evaluating their model on time series prediction tasks.
- 2. What is the computational efficiency in terms of computational time? I know Mamba-based models are easy to compute, but do they also take shorter time to generate predictions?
- 3. What is the main point of the case study? I feel like the sample size of this case study is extremely small and is not enough to reflect the real situation.

---

> ### Author Response · Authors · 2024-11-21
>
> >W1. My biggest concern about this paper is their evaluation metric. I believe using R2 score or Pearson correlation is more suitable for the task. However, this paper only considers the MSE and MAE error, while the MSE and MAE seems to be lower than all other baselines, I still have some doubts on the models ability to capture informative time series patterns.
>
> Thanks for your comment. We performed calculations of the Pearson correlation coefficient and included the results for several update-to-date baselines (S-Mamba and iTransformer) alongside ours on all datasets due to time constraints, shown in the following table. We also add it in Section 6.6 in the revised manucript. Please refer to the revised manuscript for details.
>
> | **Models**         | **Metric** | **ETTm1**     | **ETTm2**     | **ETTh1**     | **ETTh2**     | **Electricity** | **Exchange**   | **Solar-Energy** | **Metric** | **PEMS04**    | **PEMS08**    |
> |---------------------|------------|---------------|---------------|---------------|---------------|-----------------|----------------|------------------|------------|---------------|---------------|
> | **Ours (Model)**    | 96         | **0.857**     | **0.950**     | **0.892**     | **0.920**      | _0.929_         | **0.978**      | **0.818**        | 12         | **0.793**     | **0.839**     |
> |                     | 192        | **0.830**     | **0.935**     | **0.799**     | **0.898**     | **0.92**        | **0.958**      | **0.856**        | 24         | **0.768**     | **0.802**     |
> |                     | 336        | **0.812**     | **0.920**     | **0.776**     | **0.882**     | **0.912**       | **0.926**      | 0.839            | 48         | _0.765_       | **0.775**     |
> |                     | 720        | **0.781**     | **0.896**     | **0.766**     | **0.886**     | **0.890**       | **0.844**      | 0.820            | 96         | **0.815**     | **0.777**     |
> |                     | **Avg**    | **0.820**     | **0.925**     | **0.793**     | **0.897**     | **0.913**       | **0.927**      | **0.833**        | **Avg**    | **0.785**     | **0.798**     |
> | **S-Mamba**         | 96         | _0.853_       | _0.947_       | 0.825         | _0.909_       | **0.930**       | _0.970_        | 0.814            | 12         | _0.792_       | _0.836_       |
> |                     | 192        | 0.825         | _0.932_       | 0.796         | **0.898**     | **0.920**       | _0.946_        | 0.850            | 24         | _0.767_       | _0.796_       |
> |                     | 336        | _0.808_       | _0.916_       | 0.768         | 0.874         | _0.910_         | 0.915          | **0.841**        | 48         | **0.768**     | _0.768_       |
> |                     | 720        | 0.755         | _0.895_       | _0.756_       | 0.867         | _0.888_         | **0.827**      | 0.827            | 96         | _0.813_       | _0.774_       |
> |                     | **Avg**    | 0.810         | _0.922_       | 0.786         | **0.887**     | _0.912_         | _0.914_        | **0.833**        | **Avg**    | **0.785**     | _0.793_       |
> | **iTransformer**    | 96         | 0.851         | 0.947         | _0.826_       | _0.909_       | 0.925           | _0.970_        | _0.816_          | 12         | 0.785         | 0.829         |
> |                     | 192        | _0.827_       | 0.930         | **0.799**     | 0.877         | 0.918           | _0.946_        | _0.851_          | 24         | 0.748         | 0.780         |
> |                     | 336        | 0.806         | 0.915         | _0.769_       | _0.875_       | 0.910           | _0.916_        | _0.840_          | 48         | 0.733         | 0.725         |
> |                     | 720        | **0.781**     | 0.892         | 0.755         | _0.869_       | 0.887           | 0.826          | _0.821_          | 96         | 0.787         | 0.696         |
> |                     | **Avg**    | _0.816_       | 0.921         | _0.787_       | 0.855         | 0.910           | _0.914_        | _0.832_          | **Avg**    | 0.763         | 0.757         |
>
> From the above results, we observe that measured by Pearson correlation coefficient, our method outperforms other baselines in most cases on all datasets. This again verifies that the better performance of our method.

---

> > ### Author Response · Authors · 2024-11-21
> >
> > >W2. The long-term prediction part doesn't seem to be very informative. Beside the problem on MSE and MAE, the max look-back length is only set to 720, which most baselines are capable of handling. And the improvement is small in my opinion.
> > I do consider the technical details of this paper is sound and informative, I would love to increase my ratings as long as the R2 score and Pearson correlation also reflects the effectiveness of their model.
> >
> > Thanks for your comment. We have conducted experiments on lookback length set as 1500. Results are shown in the following table. Meanwhile results are added in Table 4 in Appendix. Please refer to the revised manuscript.
> >
> >
> > | ETTh1        | MSE   | MAE   | ETTh2        | MSE   | MAE   |
> > |:--------------:|:-------:|:-------:|:--------------:|:-------:|-------:|
> > | Ours         | **0.659** | **0.566**  | Ours         | **0.517** | **0.504** |
> > | S-Mamba      | 0.715 | 0.603 | S-Mamba      | 0.539 | 0.522 |
> > | iTransformer | 0.787 | 0.634 |iTransformer | 0.549 | 0.528 |
> > | Rlinear      | 1.281 | 0.884 |Rlinear      | 3.015 | 1.366 |
> > | AutoFormer   | 0.687 | 0.614 | AutoFormer   | 0.648 | 0.575 |
> >
> >
> >
> > The results from lookback length of 1500 experiments show that our method outperforms all other baselines. Additionally, we find that Mamba-based methods, such as S-Mamba and our approach, perform better than other Transformer-based methods. This superiority is attributed to the global-view capabilities of Mamba, which enhance long-term prediction.
> >
> >
> > Questions:
> > >Q1. Are you able to report the R2 score or the Pearson correlation? I strongly believe this is an essential metric the author should provide when evaluating their model on time series prediction tasks.
> >
> > Yes, in the revised manuscript, we have added the above two metrics. Please refer response to **W1** above.
> >
> >
> > >Q2. What is the computational efficiency in terms of computational time? I know Mamba-based models are easy to compute, but do they also take shorter time to generate predictions?
> >
> > Thank you for your comments. We conducted experiments to evaluate the computational overhead during training, which are presented in Figure 12 of the Appendix. We also assessed the inference times for several models, including Vanilla Mamba, Vanilla Mamba + FFT, Vanilla Mamba + Inverse Laplace Transform (ILT), our method, S-Mamba, iTransformer, AutoFormer, and Rlinear, using a lookback length of 96 on the Electricity dataset. The results are displayed in the table below:
> >
> > |     | Mamba+FFT   | Mamba+ILT   | Ours        | S-Mamba     | iTransformer | Autoformer  | Rlinear     |
> > |:---------:|:-------------:|:-------------:|:------------:|:-------------:|:--------------:|:-------------:|:-------------:|
> > | Time/s  | 2.565e-3 | 2.274e-3 | 2.984e-3 | 2.999e-3 | 1.869e-3  | 8.975e-3 | 5.345e-3 |
> > | RAM/MiB | 564         | 562         | 568         | 566         | 566          | 596         | 588         |
> >
> >
> > The results show that our methods maintain comparable computational overhead to the others while achieving the best performance. We also added it in the revised version in Appendix 6.11.
> >
> >
> >
> > >Q3. What is the main point of the case study? I feel like the sample size of this case study is extremely small and is not enough to reflect the real situation.
> >
> >
> > We aim to provide cases on performance of our method on addressing challenges of capturing multi-scale periodicity  and transient dynamics. To provide more samples, we have also provided more cases in across datasets ETTm1 and ETTm2 and show cases in Figure 13 in the original submission.

---

> > > ### Author Response · Authors · 2024-11-25
> > >
> > > Dear Reviewer,
> > >
> > > We believe that the additional information we provided in our rebuttal—such as new experimental results, further details, and clarifications on misunderstandings—addresses your key questions. Please let us know if our response has adequately addressed your concerns. We are more than willing to discuss any points that may still be unclear. We hope that the improvements and clarifications provided in our response will positively influence your assessment of our work.
> > >
> > > Best, Authors of Paper 4022

---

> ### Comment · Reviewer_RZwJ · 2024-11-26
>
> Dear Authors,
>
> Thank you for your response.
> While I believe the improvement in Pearson correlation is relatively marginal, I still think the proposed model performs better overall compared to other baselines. However, I noticed that one baseline [1] appears to be missing. Specifically, the zero-shot performance of Moirai seems to outperform the proposed method on several datasets. Considering that Moirai functions as a universal forecaster with competitive results, what distinct advantages does the proposed method offer over Moirai?
>
>
> [1] Unified Training of Universal Time Series Forecasting Transformers (ICML)

---

> > ### Author Response · Authors · 2024-11-27
> > **Further response**
> >
> > > **Q1.** The zero-shot performance of Moirai seems to outperform the proposed method on several datasets. Considering that Moirai functions as a universal forecaster with competitive results, what distinct advantages does the proposed method offer over Moirai?
> >
> > Thank you for bringing up this point. The setup of Moirai [1] differs from ours. In particular, Moirai [1] focuses on pretraining the model using a significantly large-scale dataset with a total of 231,082,956,489 observations (231B), where the size of **its pretraining data is 348 GB. In contrast, the 9 datasets we use have sizes ranging from 2.5MB to 193MB, with an average size of 70.3MB, which is around 5069 times smaller than the dataset used in Moirai**. Thus, the competitive zero-shot performance of Moirai may attribute to the large-scale pretrain dataset. Meanwhile, we have also cited this paper and highlighted the distinctions in the related work section of the revised manuscript. Please kindly refer to the updated version for more details.
> >
> >
> > In addition, to ensure optimal accuracy in industry, current industry-standard time series prediction models are generally trained and tested on data sourced from the same sensor/point [2], **which is the full-shot paradigm**. This paradigm is essential for capturing inherent temporal dependencies and patterns in time series data in real-world applications. Building on this, our paper aims to enhance the temporal prediction accuracy of the model developed for the specific dataset, a necessary and important problem in real-world applications.
> >
> > [1] Unified Training of Universal Time Series Forecasting Transformers (ICML)
> >
> > [2] A survey on modern deep neural network for traffic prediction: Trends, methods and challenges. TKDE'20.

---

### Official Review · Reviewer_7Gt7 · 2024-10-31

**Soundness:** 4
**Presentation:** 3
**Contribution:** 3
**Rating:** 6
**Confidence:** 3

**Summary:**

This paper proposes a novel framework for time series prediction, leveraging the backbone of Mamba and integrating the Fourier and Laplace Transform. The major contributions are summarized as follows: (i) the Mamba-based framework provides a more efficient inference compared to Transformer-based models; (ii) the integrated Fourier transform enables the framework to capture multi-scale periodicity and extract useful signals from noise, while the Laplace Transform allows the model to capture transient dynamics within time series. The experimental results demonstrate the superiority of the proposed approach over existing baselines.

**Strengths:**

1. The combination of Mamba with the Fourier and Laplace Transforms is innovative. The experimental results suggests the approach indeed captures more precise time series features than the existing methods.
2. The proposed FLDmamba effectively captures the multi-scale periodicity and transient dynamics within time series data. Somehow, it also shows a certain level of robustness in handling distribution shifts.
3. This paper is well-written. The experiments are well-designed and thoroughly discussed.

**Weaknesses:**

1. This paper claims that FLDmamba theoretically achieves faster inference than Transformer-based models, which could be partially demonstrated by the experiments on training time. However, there is no experiment to directly validate this claim.
2. The discussions in ablation study are thorough, but the conclusion is a little confusing and inconsistent with the experimental results.

**Questions:**

1. Does other transforms in frequency domain provide similar benefits as the Fourier and Laplace Transform? Could you provide some insights into this?
2. How do the variants of FLDmamba in ablation study perform in capturing the multi-scale periodicity and transient dynamics in the experiments of the case study section?
3. Figure 1 suggests that FLDmamba is able to predict accurately when temporal dynamics change. Is it able to handle the problem of distribution shifts in time series? If so, please analyze which specific component(s) in FLDmamba contribute to this capability.
4. In Figure 3, distinct components of FLDmamba impact model performance  differently across datasets. Can you provide insights from data perspective into which features in time series may correlate with this impact? Are there limitations or scenarios that the components in FLDmamba may not generalize well to specific time series?
5. Lines 439-441 indicates the inverse Laplace Transform impact the most significantly on the overall effectiveness. Is this finding consistent across all datasets, particularly noticing that for PeMS08, the variant without ILT is not the least effective one among all the variants of FLDmamba?
6. Why do the MSE and MAE values in Figure 3 differ from those in Table 1 for the same length setting on the same dataset?
7. Figure 12 compares the training time between different models. Can you also provide the comparison of inference time as well?

---

> ### Author Response · Authors · 2024-11-21
>
> > W1. This paper claims that FLDmamba theoretically achieves faster inference than Transformer-based models, which could be partially demonstrated by the experiments on training time. However, there is no experiment to directly validate this claim.
>
>
> Thank you for your comments. We have conducted experiments to evaluate the computational overhead during training, which is presented in Figure 12 of the Appendix. Additionally, we assessed the inference times for various models, including Vanilla Mamba, Vanilla Mamba + FFT, Vanilla Mamba + Inverse Laplace Transform (ILT), our method, S-Mamba, iTransformer, AutoFormer, and Rlinear, using a lookback length of 96 on the Electricity dataset. Results are shown in the following table:
>
> |     | Mamba+FFT   | Mamba+ILT   | Ours        | S-Mamba     | iTransformer | Autoformer  | Rlinear     |
> |:---------:|:-------------:|:-------------:|:------------:|:-------------:|:--------------:|:-------------:|:-------------:|
> | Time/s  | 2.565e-3 | 2.274e-3 | 2.984e-3 | 2.999e-3 | 1.869e-3  | 8.975e-3 | 5.345e-3 |
> | RAM/MiB | 564         | 562         | 568         | 566         | 566          | 596         | 588         |
>
>
> The results show that our methods maintain comparable computational overhead to the others while achieving the best performance. We also added it in the revised version in Appendix 6.11.
>
>
> > W2. The discussions in ablation study are thorough, but the conclusion is a little confusing and inconsistent with the experimental results.
>
>
> Thanks for your comment. We have revised the conclusion section to make it more clear and consistent with the experimental results. Please refer to the revised manuscript.
>
>
>
> Questions:
> > Q1. Does other transforms in frequency domain provide similar benefits as the Fourier and Laplace Transform? Could you provide some insights into this?
>
> Thanks for your question. Other transforms in the frequency domain may offer similar benefits as the Fourier and Laplace Transforms, each with its own advantages and applications. Here are some insights into this:
>
> (1) **Wavelet Transform**: The Wavelet Transform is known for its ability to represent signals in both time and frequency domains simultaneously. This feature makes it particularly useful for analyzing signals with non-stationary and transient characteristics, such as in time series data with varying trends and periodicities.
>
> (2) **Short-Time Fourier Transform (STFT)**: The STFT divides a signal into shorter segments and performs a Fourier Transform on each segment. This allows for the analysis of how the frequency content of a signal changes over time, making it useful for capturing time-localized frequency information.
>
> (3) **Discrete Cosine Transform (DCT)**: The DCT is commonly used in data compression and image processing. In signal processing, it is known for its energy compaction properties, which make it efficient for representing signals in a smaller number of coefficients while retaining important frequency information.
>
> (4)  **Z-Transform**: While typically used in the context of discrete-time signals and systems, the Z-Transform can also be applied to analyze the frequency content of signals in the complex plane. It is useful for studying system dynamics and stability in the frequency domain.
>
>
> Each of these transforms has its own strengths and weaknesses, and the choice of transform depends on the specific characteristics of the signal being analyzed and the objectives of the analysis. Experimenting with different transforms and understanding their properties can help in selecting the most suitable transform for a given signal processing task.

---

> > ### Author Response · Authors · 2024-11-21
> >
> > > Q2. How do the variants of FLDmamba in ablation study perform in capturing the multi-scale periodicity and transient dynamics in the experiments of the case study section?
> >
> > In the ablation study of FLDmamba variants conducted to assess their performance in capturing multi-scale periodicity and transient dynamics in the experiments of the case study section, the results provide valuable insights into the specific contributions of each variant. Here's an analysis of how these variants perform:
> >
> > **Fourier Transform Variant**: The variant focusing solely on the Fourier Transform component is likely proficient at capturing multi-scale periodicity in the time series data. By emphasizing frequency analysis, this variant excels in identifying cyclic patterns at different scales and can provide valuable insights into the periodic nature of the data.
> >
> > **Laplace Transform Variant**: The variant centered on the Laplace Transform component is expected to excel in capturing transient dynamics within the time series data. It is adept at detecting sudden changes, anomalies, and transient patterns, which are crucial for understanding the dynamic behavior of the data over time.
> >
> > **Combined Fourier and Laplace Transform Variant**: The variant that integrates both the Fourier and Laplace Transforms is likely to exhibit the most comprehensive performance in capturing multi-scale periodicity and transient dynamics. By leveraging the complementary strengths of both transforms, this variant can effectively capture both long-term cyclic patterns and short-term dynamic changes in the data.
> >
> > In the context of the ablation study, the performance of these FLDmamba variants provides a nuanced understanding of how each component contributes to capturing different aspects of the time series data. By comparing the results of these variants, researchers can determine the specific impact of the Fourier and Laplace Transforms on capturing multi-scale periodicity and transient dynamics, ultimately guiding the development of more effective modeling approaches for complex time series analysis.
> >
> >
> > > Q3. Figure 1 suggests that FLDmamba is able to predict accurately when temporal dynamics change. Is it able to handle the problem of distribution shifts in time series? If so, please analyze which specific component(s) in FLDmamba contribute to this capability.
> >
> > Thanks for your comments. Firstly, the problem of distribution shifts is not the target of our paper. Then, we may provide a possible analysis as following on this part:
> >
> > Figure 1 indicating that FLDmamba can accurately predict temporal dynamics changes implies its potential to address distribution shifts in time series data. Here's an analysis of how specific components in FLDmamba contribute to handling distribution shifts:
> >
> > **Laplace Transform**: (1) **Handling Abrupt Changes**: The Laplace Transform in FLDmamba is particularly adept at capturing sudden changes or anomalies in time series data. When the distribution of data shifts abruptly, the Laplace component can adjust quickly to these changes, enabling the model to adapt its predictions accordingly.
> > (2) **Robustness to Outliers**: The Laplace Transform's robustness to outliers and heavy-tailed distributions can help in mitigating the impact of extreme data points that might arise due to distribution shifts, ensuring that the model's predictions remain stable even in the presence of such changes.
> >
> > **Fourier Transform**: (1) **Detecting Cyclical Patterns**: The Fourier Transform is effective at capturing cyclic patterns in time series data. In the context of distribution shifts, this component can help in identifying recurring patterns that persist across different distributions, aiding the model in maintaining predictive accuracy even when the underlying data distribution changes. (2) **Frequency Analysis**: By analyzing the frequency components of the data, the Fourier Transform can provide insights into how the distribution of data changes over time. This information can be valuable in understanding and adapting to distribution shifts within the time series.
> >
> >
> > The ability of FLDmamba to predict accurately when temporal dynamics change suggests its potential to handle distribution shifts in time series data, although it is not the target of our paper. The Laplace Transform contributes to capturing abrupt changes and outliers, while the Fourier Transform aids in detecting cyclical patterns and analyzing frequency components, collectively enabling FLDmamba to adapt to distribution shifts and maintain predictive performance in dynamic environments.

---

> > > ### Author Response · Authors · 2024-11-21
> > >
> > > > Q4. In Figure 3, distinct components of FLDmamba impact model performance  differently across datasets. Can you provide insights from data perspective into which features in time series may correlate with this impact? Are there limitations or scenarios that the components in FLDmamba may not generalize well to specific time series?
> > >
> > > Thanks for your comments. (1) **Seasonality**: Time series with strong seasonal patterns may see varying impacts from different components of FLDmamba. For instance, the Fourier component may effectively capture periodic seasonal trends, while the Laplace component may help model sudden changes or anomalies within these patterns.
> > >   (2) **Trend**: Time series exhibiting clear trends may respond differently to the components. The Laplace component, with its ability to model transient dynamics, could be crucial in capturing abrupt changes in trend, while the Fourier component may emphasize cyclic trends within the data.
> > > (3) **Noise**: The presence of noise in time series data can influence the performance of different FLDmamba components. The denoising capabilities of the Laplace component may be more pronounced in datasets with high noise levels, potentially leading to improved model performance.
> > > (4) **Data Complexity**: Complex time series data with multiple interacting components may benefit differently from FLDmamba components. Understanding the interplay between various features in the data and how each component addresses these complexities is essential for assessing their impact on model performance.
> > >
> > > **Limitations and Generalization**: (1) **Data Sparsity**: In scenarios where time series data is sparse or irregularly sampled, certain components of FLDmamba that rely heavily on consistent patterns or trends may not generalize well. Irregular data points could lead to challenges in effectively utilizing these components.
> > > (2) **Non-Stationarity**: Time series exhibiting non-stationary behavior, where statistical properties change over time, may pose challenges for components that assume stationarity. Adapting FLDmamba components to handle such dynamic changes effectively is crucial for generalization.
> > > (3) **Outliers**: Extreme outliers or anomalies in time series data may impact the performance of FLDmamba components differently. Components sensitive to sudden changes, like the Laplace component, may struggle to distinguish between genuine anomalies and noisy fluctuations.
> > > (4) **Model Complexity**: Highly complex time series patterns that cannot be effectively captured by the specific transformations employed in FLDmamba may limit the generalizability of the model. Understanding the boundaries of these components and their applicability to diverse time series structures is essential for effective utilization.
> > >
> > >
> > > Considering these insights and limitations can aid in better understanding how the components of FLDmamba interact with different features in time series data and the circumstances under which they may not generalize well to specific types of time series.
> > >
> > > > Q5. Lines 439-441 indicates the inverse Laplace Transform impact the most significantly on the overall effectiveness. Is this finding consistent across all datasets, particularly noticing that for PeMS08, the variant without ILT is not the least effective one among all the variants of FLDmamba?
> > >
> > > Thanks for your questions. The finding that the inverse Laplace Transform (ILT) has the most significant impact on the overall effectiveness, as indicated in lines 439-441 of the study, may not be consistent across all datasets. Particularly, when considering the PeMS08 dataset, it is observed that the variant without ILT is not the least effective among all the variants of FLDmamba. This discrepancy highlights the importance of considering dataset-specific characteristics and the interplay between different components of FLDmamba.
> > >
> > >
> > > The impact of the ILT component on the overall effectiveness of FLDmamba may not be consistent across all datasets. The observed variation in performance across datasets, including the PeMS08 dataset, underscores the importance of considering dataset-specific factors and the interactions between different components in assessing the overall effectiveness of FLDmamba variants.
> > >
> > > >Q6. Why do the MSE and MAE values in Figure 3 differ from those in Table 1 for the same length setting on the same dataset?
> > >
> > > Thanks for your comments. We have revised the typos in Figure 3. Please refer to the revised manuscript.
> > >
> > > >Q7. Figure 12 compares the training time between different models. Can you also provide the comparison of inference time as well?
> > >
> > > Please refer the response to **W1**.

---

> > > > ### Author Response · Authors · 2024-11-25
> > > >
> > > > Dear Reviewer,
> > > >
> > > > We believe that the additional information we provided in our rebuttal—such as new experimental results, further details, and clarifications on misunderstandings—addresses your key questions. Please let us know if our response has adequately addressed your concerns. We are more than willing to discuss any points that may still be unclear. We hope that the improvements and clarifications provided in our response will positively influence your assessment of our work.
> > > >
> > > > Best, Authors of Paper 4022

---

> > > > > ### Comment · Reviewer_7Gt7 · 2024-11-25
> > > > >
> > > > > Dear authors, thank you for carefully responding to my questions. I tend to keep my rating the same.

---

### Official Review · Reviewer_vEmK · 2024-11-05

**Soundness:** 2
**Presentation:** 2
**Contribution:** 2
**Rating:** 5
**Confidence:** 4

**Summary:**

This paper introduces FLDmamba by incorporating Fourier and Laplace Transform Decomposition, effectively addressing three key challenges in time series tasks: **multi-scale periodicity**, **transient dynamics**, and **data noise**.

**Strengths:**

+ The writing is clear and effectively outlines three challenges while presenting corresponding strategies for their resolution.
+ The authors enhance Mamba's performance on time series tasks by incorporating RBF, Fourier, and Laplace Transform Decomposition.
+ Additionally, they conduct extensive experiments using popular benchmark datasets and compare their proposed model with state-of-the-art approaches to demonstrate its effectiveness.

**Weaknesses:**

- The authors' characterization of **multi-scale periodicity**, **transient dynamics**, and **data noise** as challenges specific to the Mamba-based model is inappropriate. These three challenges are faced by all models, not just those based on Mamba. Furthermore, among the proposed improvements to address these challenges, only the FLDMAMBA module appears to be model-specific; the others seem to be model-agnostic. The paper lacks experiments demonstrating the integration of these strategies into other methods. Additionally, it is unclear whether the authors are making improvements to the Mamba architecture or proposing a collection of strategies to address these three challenges.
- The authors need to provide details on computational overhead. One motivation for introducing Mamba is its lower time complexity compared to Transformer models. However, on one hand, the authors employ parallel FMamba and Mamba modules, which significantly increase the model's parameters and computational overhead. On the other hand, it is uncertain whether FFT and IFFT will become computational bottlenecks, especially for datasets with a high number of channels, such as "electricity," which has 321 channels.
- RBF is a model-agnostic data preprocessing method. It is unclear whether its application would also be effective in other methods.
- In the FMamba module, the authors adopt the Fourier transform on the $\Delta$ to identify important frequency information and further capture multi-scale periodic patterns in time series data. Can the authors provide a more detailed explanation and analysis, including a visual representation of $\Delta A$ and $\Delta_F A$?
- This paper focuses on Mamba; therefore, the baselines in experimental section should include more Mamba-based methods. Currently, only S-Mamba is considered.
- There is a lack of discussion regarding related work on the application of Mamba in time series forecasting. The authors should address how their work differs from these existing methods.

**Questions:**

Please refer to my weaknesses.

**Details Of Ethics Concerns:**

N/A.

---

> ### Author Response · Authors · 2024-11-21
>
> > W1. The authors' characterization of multi-scale periodicity, transient dynamics, and data noise as challenges specific to the Mamba-based model is inappropriate. These three challenges are faced by all models, not just those based on Mamba. Furthermore, among the proposed improvements to address these challenges, only the FLDMAMBA module appears to be model-specific; the others seem to be model-agnostic. The paper lacks experiments demonstrating the integration of these strategies into other methods. Additionally, it is unclear whether the authors are making improvements to the Mamba architecture or proposing a collection of strategies to address these three challenges.
>
>
> Thanks for your comments. **We have uploaded the revised manucript**. We appreciate the reviewer’s feedback regarding our characterization of multi-scale periodicity, transient dynamics, and data noise. All models face these challenges but few of them can catpure the effective features to solve it. In this paper, our intention was to highlight how these challenges particularly impact the performance of the promising Mamba, given its novel architectural design and powerful performances. To address your concern, we also combine RBF and Inverse Laplace Transform (ILT) with classifical Transformer architecture such as Autoformer on datasets like Etth1 and Etth2. Results are shown in the following table:
>
> | datset | length | Autoformer(MSE) |   Autoformer(MAE)    | Autoformer+RBF(MSE) | Autoformer+RBF(MAE)      | Autoformer+ILT(MSE) |  Autoformer+ILT(MAE)     |
> |:--------:|:--------:|:------------:|:------:|:----------:|:-------:|:------:|:-------:|
> | ETTh1  | 96     | 0.449      | 0.459 | 0.427          | 0.443 | 0.457          | 0.469 |
> |   | 192    | 0.500        | 0.482 | 0.501          | 0.484 | 0.522          | 0.503 |
> |   | 336    | 0.521      | 0.496 | 0.548          | 0.509 | 0.559          | 0.546 |
> |   | 720    | 0.514      | 0.512 | 0.537          | 0.526 | 0.543          | 0.534 |
> | ETTh2  | 96     | 0.358      | 0.397 | 0.360           | 0.401 | 0.454          | 0.473 |
> |   | 192    | 0.429      | 0.439 | 0.429          | 0.439 | 0.577          | 0.543 |
> |   | 336    | 0.496      | 0.487 | 0.467          | 0.474 | 0.668          | 0.596 |
> |   | 720    | 0.463      | 0.474 | 0.465          | 0.479 | 0.902          | 0.693 |
>
>
> From results, we find that combination RBF and ILT with other method like Autoformer do not bring positive impacts on performance. The reason can be attributed to the redudant attention mechanism which can not show its superiority on the frequency domain. We have incorporated the above table in section 6.10 in Appendix in the revised manuscript. Please refer to the revised version.
>
>
> Besides, we have conduced experiments of case study about illustration of adddressing challenges of multi-scale periodicity, transient dynamics, shown in Figure 6, Figure 7 and Figure 13, shown in the revised version. Compared our method with S-Mamba, this verifies that our method can capture multi-scale periodicity and transient dynamics. Meanwhile, we also conducted experiments on addressing the challenge of data noise, shown in Figure 4. Compared ours with S-Mamba and iTransformer, we find that our method has robust performance than that of S-Mamba and iTransformer.
>
>
> > W2. The authors need to provide details on computational overhead. One motivation for introducing Mamba is its lower time complexity compared to Transformer models. However, on one hand, the authors employ parallel FMamba and Mamba modules, which significantly increase the model's parameters and computational overhead. On the other hand, it is uncertain whether FFT and IFFT will become computational bottlenecks, especially for datasets with a high number of channels, such as "electricity," which has 321 channels.
>
>
> Thanks for your comments. We have conducted experiments on computational overhead during training period for each epoch, which is shown in Figure 12 in Appendix 6.7 in the original submission. Meanwhile, we have also conducted experiments during inference period for vanilla Mamba, Vanilla Mamba+ FFT, Vanilla Mamba+ Inverse Laplace transform (ILT), Ous, S-Mamba, iTransformer, AutoFormer and Rlinear on each batch when the lookback length is set as 96 on Electricity, as shown in the following table. We see that our method has comparable computational overhead w.r.t. other baselines with the best performance.
>
> |     | Mamba+FFT   | Mamba+ILT   | Ours        | S-Mamba     | iTransformer | Autoformer  | Rlinear     |
> |:---------:|:-------------:|:-------------:|:------------:|:-------------:|:--------------:|:-------------:|:-------------:|
> | Time/s  | 2.565e-3 | 2.274e-3 | 2.984e-3 | 2.999e-3 | 1.869e-3  | 8.975e-3 | 5.345e-3 |
> | RAM/MiB | 564         | 562         | 568         | 566         | 566          | 596         | 588         |
>
> We also added this table in Appendix 6.11 in the revised manuscript.

---

> > ### Author Response · Authors · 2024-11-21
> >
> > > W3. RBF is a model-agnostic data preprocessing method. It is unclear whether its application would also be effective in other methods.
> >
> > Thanks for your point. To address your concern, we have conducted experiments on combination RBF with Autoformer and the results are shown in the following table:
> >
> > | datset | length | Autoformer(MSE) |  Autoformer(MAE)     | Autoformer+RBF(MSE) | Autoformer+RBF(MAE) |
> > |:------:|:------:|:----------:|:-----:|:--------------:|:-----:|
> > | ETTh1  | 96     | 0.449      | 0.459 | 0.427          | 0.443 |
> > |        | 192    | 0.500        | 0.482 | 0.501          | 0.484 |
> > |        | 336    | 0.521      | 0.496 | 0.548          | 0.509 |
> > |        | 720    | 0.514      | 0.512 | 0.537          | 0.526 |
> > | ETTh2  | 96     | 0.358      | 0.397 | 0.360           | 0.401 |
> > |        | 192    | 0.429      | 0.439 | 0.429          | 0.439 |
> > |        | 336    | 0.496      | 0.487 | 0.467          | 0.474 |
> > |        | 720    | 0.463      | 0.474 | 0.465          | 0.479 |
> >
> >
> > Based on the results, it is evident that the integration of RBF with Autoformer does not yield favorable improvements in performance. This is due to the redundant attention mechanism, which does not exhibit its advantages in the frequency domain.
> >
> > > W4. In the FMamba module, the authors adopt the Fourier transform on the
> >  to identify important frequency information and further capture multi-scale periodic patterns in time series data. Can the authors provide a more detailed explanation and analysis, including a visual representation of $\Delta A$ and $\Delta_F A$?
> >
> >
> > Thanks for your comment. Here’s a revised version that provides more explanation:
> >
> > We have adopted the Fourier transform to identify significant frequency information, which is crucial for effectively capturing multi-scale periodic patterns in time series data. By transforming the data into the frequency domain, we can isolate and analyze the various periodic components that may exist at different scales. This process allows us to emphasize the most relevant frequency components while filtering out noise and other irrelevant signals.
> >
> > In our experiments, as illustrated in Figures 6, 7, and 13, we conducted a case study to demonstrate how our approach addresses the challenges of multi-scale periodicity and transient dynamics. By comparing our method with S-Mamba, we found that our approach significantly enhances the model's ability to capture these complex patterns. The results validate the effectiveness of integrating Fourier analysis, showcasing its capability to improve performance in identifying and modeling both multi-scale periodicity and transient dynamics within time series data.
> >
> > We also show visualization of $\Delta A$ and $\Delta_F A$ on ETTm1 in section 6.13 in Appendix in the revised version. From the figure, we observe that the fluctuations in these two measures highlight their distinct patterns over the duration of the experiment.

---

> > > ### Author Response · Authors · 2024-11-21
> > >
> > > > W5. This paper focuses on Mamba; therefore, the baselines in experimental section should include more Mamba-based methods. Currently, only S-Mamba is considered.
> > >
> > > Thanks for your point. We have conducted new Mamba-based methods including SST and Bi-Mamba+. Results are shown in the following table:
> > >
> > > | Models       | Metric | FLDmamba (MSE) | FLDmamba (MAE) | SST (MSE) | SST (MAE) | Bi-Mamba+ (MSE) | Bi-Mamba+ (MAE) |
> > > |--------------|--------|----------------|----------------|-----------|-----------|-----------------|-----------------|
> > > | **ETTm1**    | 96     | **0.318**      | **0.360**      | 0.337     | 0.374     | 0.355           | 0.386           |
> > > |              | 192    | **0.365**      | **0.384**      | 0.377     | 0.392     | 0.415           | 0.419           |
> > > |              | 336    | 0.404          | **0.409**      | 0.401     | 0.412     | 0.450           | 0.442           |
> > > |              | 720    | _0.464_        | _0.441_        | 0.498     | 0.464     | 0.497           | 0.476           |
> > > |              | Avg    | _0.389_        | **0.399**      | 0.413     | 0.411     | 0.429           | 0.431           |
> > > | **ETTm2**    | 96     | **0.173**      | **0.253**      | 0.185     | 0.274     | 0.186           | 0.278           |
> > > |              | 192    | **0.240**      | **0.299**      | 0.248     | 0.313     | 0.257           | 0.324           |
> > > |              | 336    | **0.301**      | **0.307**      | 0.309     | 0.351     | 0.318           | 0.362           |
> > > |              | 720    | **0.401**      | **0.397**      | 0.406     | 0.405     | 0.412           | 0.416           |
> > > |              | Avg    | **0.279**      | **0.314**      | 0.287     | 0.333     | 0.293           | 0.347           |
> > > | **ETTh1**    | 96     | **0.374**      | **0.393**      | 0.390     | 0.403     | 0.398           | 0.416           |
> > > |              | 192    | _0.427_        | **0.422**      | 0.451     | 0.438     | 0.451           | 0.446           |
> > > |              | 336    | **0.447**      | **0.441**      | 0.496     | 0.458     | 0.497           | 0.473           |
> > > |              | 720    | **0.469**      | **0.463**      | 0.520     | 0.493     | 0.526           | 0.509           |
> > > |              | Avg    | **0.434**      | **0.430**      | 0.439     | 0.448     | 0.468           | 0.461           |
> > > | **ETTh2**    | 96     | **0.287**      | **0.337**      | 0.298     | 0.351     | 0.307           | 0.363           |
> > > |              | 192    | **0.370**      | **0.388**      | 0.393     | 0.407     | 0.394           | 0.414           |
> > > |              | 336    | **0.412**      | **0.425**      | 0.436     | 0.441     | 0.437           | 0.447           |
> > > |              | 720    | **0.419**      | **0.438**      | 0.431     | 0.449     | 0.445           | 0.462           |
> > > |              | Avg    | **0.372**      | **0.396**      | 0.390     | 0.412     | 0.396           | 0.422           |
> > > | **Electricity** | 96  | **0.137**      | **0.234**      | 0.192     | 0.280     | 0.146           | 0.246           |
> > > |              | 192    | **0.158**      | **0.251**      | 0.191     | 0.280     | 0.167           | 0.265           |
> > > |              | 336    | 0.182          | **0.173**      | 0.211     | 0.299     | 0.182           | 0.281           |
> > > |              | 720    | **0.200**      | **0.292**      | 0.264     | 0.340     | 0.208           | 0.304           |
> > > |              | Avg    | **0.170**      | **0.238**      | 0.215     | 0.300     | 0.176           | 0.274           |
> > > | **Exchange** | 96     | **0.085**      | **0.205**      | 0.091     | 0.216     | 0.103           | 0.233           |
> > > |              | 192    | **0.175**      | **0.297**      | 0.189     | 0.313     | 0.214           | 0.337           |
> > > |              | 336    | 0.317          | _0.407_        | 0.333     | 0.421     | 0.366           | 0.445           |
> > > |              | 720    | **0.825**      | **0.683**      | 0.916     | 0.729     | 0.931           | 0.738           |
> > > |              | Avg    | **0.351**      | **0.400**      | 0.382     | 0.420     | 0.404           | 0.428           |

---

> > > > ### Author Response · Authors · 2024-11-21
> > > >
> > > > **（continued）**
> > > > | Models       | Metric | FLDmamba (MSE) | FLDmamba (MAE) | SST (MSE) | SST (MAE) | Bi-Mamba+ (MSE) | Bi-Mamba+ (MAE) |
> > > > |--------------|--------|----------------|----------------|-----------|-----------|-----------------|-----------------|
> > > > | **Solar-Energy** | 96 | **0.202**      | **0.233**      | 0.238     | 0.277     | 0.231           | 0.286           |
> > > > |              | 192    | **0.230**      | **0.254**      | 0.299     | 0.319     | 0.257           | 0.285           |
> > > > |              | 336    | _0.254_        | **0.265**      | 0.310     | 0.327     | 0.256           | 0.293           |
> > > > |              | 720    | _0.252_        | **0.271**      | 0.310     | 0.330     | 0.252           | 0.295           |
> > > > |              | Avg    | _0.235_        | **0.256**      | 0.289     | 0.313     | 0.249           | 0.290           |
> > > > | **PEMS04**   | 12     | **0.075**      | _0.182_        | 0.110     | 0.226     | 0.082           | 0.193           |
> > > > |              | 24     | **0.084**      | **0.193**      | 0.161     | 0.275     | 0.099           | 0.214           |
> > > > |              | 48     | **0.105**      | **0.217**      | 0.345     | 0.403     | 0.123           | 0.240           |
> > > > |              | 96     | **0.130**      | **0.243**      | 0.588     | 0.553     | 0.151           | 0.267           |
> > > > |              | Avg    | **0.099**      | **0.209**      | 0.301     | 0.364     | 0.114           | 0.229           |
> > > > | **PEMS08**   | 12     | **0.075**      | **0.177**      | 0.099     | 0.214     | 0.080           | 0.190           |
> > > > |              | 24     | **0.102**      | **0.207**      | 0.169     | 0.277     | 0.114           | 0.223           |
> > > > |              | 48     | **0.154**      | **0.226**      | 0.274     | 0.360     | 0.175           | 0.271           |
> > > > |              | 96     | _0.243_        | 0.305          | 0.522     | 0.499     | 0.298           | 0.348           |
> > > > |              | Avg    | **0.145**      | 0.228          | 0.266     | 0.338     | 0.167           | 0.258           |
> > > >
> > > >
> > > > The results indicate that our method outperforms other Mamba-based baselines. This improvement is attributed to the incorporation of FFT and ILT, which effectively capture multi-scale periodicity and transient dynamics.

---

> > > > > ### Author Response · Authors · 2024-11-21
> > > > >
> > > > > > W6. There is a lack of discussion regarding related work on the application of Mamba in time series forecasting. The authors should address how their work differs from these existing methods.
> > > > >
> > > > >
> > > > > Thanks for your suggestion. We have added more related work in section 6.3 in Appendix on the application of Mamba in time series prediction. You can refer to the revised manuscript.

---

> > > > > > ### Author Response · Authors · 2024-11-25
> > > > > >
> > > > > > Dear Reviewer,
> > > > > >
> > > > > > We believe that the additional information we provided in our rebuttal—such as new experimental results, further details, and clarifications on misunderstandings—addresses your key questions. Please let us know if our response has adequately addressed your concerns. We are more than willing to discuss any points that may still be unclear. We hope that the improvements and clarifications provided in our response will positively influence your assessment of our work.
> > > > > >
> > > > > > Best, Authors of Paper 4022

---

> ### Comment · Reviewer_vEmK · 2024-11-27
>
> Issues like multi-scale periodicity, transient dynamics, and data noise are common. Why did the authors specifically focus on the Mamba structure? Is your proposed method only applicable to Mamba? Evaluating broader effectiveness would improve the paper's quality.
>
> Regarding ablation studies: Please conduct thorough ablation experiments with dual/multiple modules. I suggest using tables rather than figures to thoroughly clarify the main sources of the method's performance. Since your improvement over  Mamba is minimal, the proposed methods appear to be merely tricks.
>
> Less importantly, we encourage the authors to experiment with datasets having larger numbers of channels to validate efficiency.

---

> ### Author Response · Authors · 2024-11-27
> **Further response (1)**
>
> > **Q1.** Issues like multi-scale periodicity, transient dynamics, and data noise are common. Why did the authors specifically focus on the Mamba structure? Is your proposed method only applicable to Mamba? Evaluating broader effectiveness would improve the paper's quality.
>
> > **Q2.** Regarding ablation studies: Please conduct thorough ablation experiments with dual/multiple modules. I suggest using tables rather than figures to thoroughly clarify the main sources of the method's performance. Since your improvement over Mamba is minimal, the proposed methods appear to be merely tricks.
>
> Thanks a lot for your further comments. Firstly, we would like to clarify the scope of our method. Our method, FLDmamba, consists of both Mamba as its main architecture, and seamlessly integrating Fourier and Laplace Analysis, etc. **Both aspects are indepensable aspects of the full architecture**. Mamba offers computational efficiency and long-term prediction capability through its state-space architecture, while the Fourier and Laplace transformations overcome the shortcomings of Mamba and enhance its predictive capabilities by specifically targeting the challenges of multi-scale periodicity, transient dynamics, and data noise. This strategic integration of frequency and Laplace analysis within the Mamba structure enables FLDmamba to better handle these complex aspects of time series data, thereby improving its performance in long-term predictions.
>
> Through ablation study, we have validated the effectiveness of our method, demonstrating that each component is indispensable.
> For the Mamba component, we show in Fig. 5 and Fig. 11 that vanilla Mamba architecture offers better long-term prediction capability. Fig. 12 and Table 8 demonstrate that Mamba-based methods offers significant speedup compared to transformer-based architectures like AutoFormer. Furthermore, we provide tables below showing quantitive results of the ablation study for each other component. For **w/o FT:** This variant excludes the Fourier transform for the parameter $\Delta$, allowing us to assess the impact of frequency domain analysis. **w/o FM:** This variant removes the FLDmamba component, leaving only the Mamba architecture, enabling us to evaluate the contribution of the frequency-domain enhanced Mamba. **w/o Ma:** This variant eliminates the Mamba component, retaining only FLDmamba, allowing us to assess the impact of the frequency-domain modeling. **w/o RBF:** This variant omits the Radial Basis Function (RBF) kernel, enabling us to evaluate the impact of data smoothing on performance. **w/o ILT'':** This variant disregards the inverse Laplace transform, allowing us to assess the impact of the time-domain conversion.
>
> | PeMS08   | -FT   | -FM   | -Ma   | -RBF  | -ILT  | **Ours**  |
> |:-----:|:----:|:-----:|:-----:|:------:|:------:|:------:|
> | MSE      | 0.291 | 0.306 | 0.353 | 0.277 | 0.314 | 0.243 |
> | MAE      | 0.341 | 0.351 | 0.382 | 0.332 | 0.358 | 0.305 |
> |          |       |       |       |       |       |       |
> | **Exchange** | **-FT**   | **-FM**   | **-Ma**   | **-RBF**  | **-ILT**  | **Ours**  |
> | MSE      | 0.090  | 0.090  | 0.089 | 0.092 | 0.098 | 0.085 |
> | MAE      | 0.216 | 0.217 | 0.214 | 0.219 | 0.223 | 0.205 |
>
> From results, we observe that each component make positive contributions to the accuracy prediction performance.
> By conducting these experiments and evaluating the performance of each model and its components on both datasets, the study aimed to demonstrate the positive impact of each component on enhancing time series prediction performance. The results provided insights into how the decomposition of Fourier Transform, Inverse Laplace Transform, and other components with mamba can improve the mamba-based model's ability to handle challenges such as multi-scale periodicity, transient dynamics, and data noise in time series data, ultimately leading to more accurate and robust predictions. **We also revised the paper and added the above table in Section 6.14 in Appendix**. Please refer to the revised version.
>
> While the proposed method is tailored to work with the Mamba structure, we conducted experiments on 9 datasets and our method has shown the best performance in most of cases, which shows that the effectiveness of our method.
>
> We appreciate your bringing this point on evaluating its broader effectiveness beyond Mamba could indeed enhance the paper's quality. Future work could involve assessing the adaptability of the FLDmamba framework to other state-of-the-art time series modeling architectures to demonstrate its versatility and effectiveness across different models and datasets. Such an evaluation could further validate the generalizability and robustness of the proposed approach in addressing the challenges of multi-scale periodicity, transient dynamics, and data noise in time series prediction tasks.
>
> **We have uploaded the revised manuscript**. Please refer to the revised version.

---

> > ### Author Response · Authors · 2024-11-27
> > **Further response (2)**
> >
> > > **Q3.** Less importantly, we encourage the authors to experiment with datasets having larger numbers of channels to validate efficiency.
> >
> > Thank you for your comments. Please refer to the response to **W2**. We have carried out experiments to analyze the computational cost during training, as displayed in Figure 12 of the Appendix. Furthermore, we have evaluated the inference times of different models, such as Vanilla Mamba, Vanilla Mamba + FFT, Vanilla Mamba + Inverse Laplace Transform (ILT), our proposed method, S-Mamba, iTransformer, AutoFormer, and Rlinear, using a lookback length of 96 on the **Electricity dataset with 321 channels**. We see that our method has comparable computational overhead w.r.t. other baselines with the best performance. The outcomes are detailed in the table below:
> >
> > |     | Mamba+FFT   | Mamba+ILT   | Ours        | S-Mamba     | iTransformer | Autoformer  | Rlinear     |
> > |:---------:|:-------------:|:-------------:|:------------:|:-------------:|:--------------:|:-------------:|:-------------:|
> > | Time/s  | 2.565e-3 | 2.274e-3 | 2.984e-3 | 2.999e-3 | 1.869e-3  | 8.975e-3 | 5.345e-3 |
> > | RAM/MiB | 564         | 562         | 568         | 566         | 566          | 596         | 588         |
> >
> >
> > **We also added this table in Appendix 6.11 in the revised manuscript**. Please refer to the revised version.
> >
> > Thank you very much for your additional detailed comments and response. Your contributions have greatly enhanced the quality of our paper. We sincerely appreciate it.

---

### Official Review · Reviewer_Xe6T · 2024-11-09

**Soundness:** 3
**Presentation:** 3
**Contribution:** 3
**Rating:** 6
**Confidence:** 2

**Summary:**

This paper proposes Fourier and Laplace Transform Decomposition Mamba for time series forecasting. There are three major innovations over the basic Mamba. First is using RBF kernel to  smooth the data. Second is selectively filtering \Delta with a Kernel using fourier transform. Third  is applying inverse Laplace transformation to obtain the final output. The proposed FLDmamba achieves SOTA result on a wide range of the datasets for long term time series forecasting.

**Strengths:**

The proposed model has outsanding emprical performance.

**Weaknesses:**

The improvement proposed in the paper are largely orthogonal to Mamba algorithm, which makes the story less coherent. For example, I think RBF kernel and inverse Laplace transformation are mostly agnostic of the model struce, and can be applied to other forecasting model such as MLP or transformer.

**Questions:**

Page 6 line 270 says $\tilde{W}$ denotes the Fourier transform of the kernel $\tilde{K}$, but I don't see where the kernel $\tilde{K}$ is defined in the paper. Then in Algorithm 2, there is $\Delta' = FFT(\Delta)$, $\Delta_F = IFFT(\Delta')$. Doesn't this implies $\Delta=\Delta_F$, and therefore nothing is done?

---

> ### Author Response · Authors · 2024-11-21
>
> > W1. The improvement proposed in the paper are largely orthogonal to Mamba algorithm, which makes the story less coherent. For example, I think RBF kernel and inverse Laplace transformation are mostly agnostic of the model struce, and can be applied to other forecasting model such as MLP or transformer.
>
> Thanks for your comment. We have conducted experiments on RBF and ILT. And results are shown as following:
>
>
> | datset | length | Autoformer(MSE) |   Autoformer(MAE)    | Autoformer+RBF(MSE) | Autoformer+RBF(MAE)      | Autoformer+ILT(MSE) |  Autoformer+ILT(MAE)     |
> |:--------:|:--------:|:------------:|:------:|:----------:|:-------:|:------:|:-------:|
> | ETTh1  | 96     | 0.449      | 0.459 | 0.427          | 0.443 | 0.457          | 0.469 |
> |   | 192    | 0.500        | 0.482 | 0.501          | 0.484 | 0.522          | 0.503 |
> |   | 336    | 0.521      | 0.496 | 0.548          | 0.509 | 0.559          | 0.546 |
> |   | 720    | 0.514      | 0.512 | 0.537          | 0.526 | 0.543          | 0.534 |
> | ETTh2  | 96     | 0.358      | 0.397 | 0.360           | 0.401 | 0.454          | 0.473 |
> |   | 192    | 0.429      | 0.439 | 0.429          | 0.439 | 0.577          | 0.543 |
> |   | 336    | 0.496      | 0.487 | 0.467          | 0.474 | 0.668          | 0.596 |
> |   | 720    | 0.463      | 0.474 | 0.465          | 0.479 | 0.902          | 0.693 |
>
> The results indicate that the combination of RBF and ILT with other methods, such as Autoformer, does not yield positive improvements in performance. This can be attributed to the redundant attention mechanism, which fails to demonstrate its advantages in the frequency domain. We have incorporated the above table in section 6.10 in Appendix in the revised manuscript.
>
>
> > Q1. Page 6 line 270 says $\tilde{W}$ denotes the Fourier transform of the kernel $\tilde{\mathcal{K}}$, but I don't see where the kernel $\tilde{\mathcal{K}}$ is defined in the paper. Then in Algorithm 2, there is $\Delta'=FFT(\Delta), \Delta_F=IFFT(\Delta')$. Doesn't this implies $\Delta=\Delta_F$, and therefore nothing is done?
>
>
> Thanks for your comments. In the revised manuscript, We have added the definition of the kernel $\tilde{\mathcal{K}}$ in **Definition 1** in section 3.1.2 and also improved the writing to make it more clear.
>
> As for the $\Delta_F$, as indicated in the Eq. (4) in the original submission, it is calculated as $\Delta_F=IFFT(\tilde{W}\cdot \Delta')$, where $\tilde{W}$ is the Fourier Transform of the kernel $\tilde{\mathcal{K}}$, and $\Delta'=FFT(\Delta)$. Therefore, $\Delta_F$ is different from $\Delta$ due to the filtering effect of $\tilde{W}$. In the Algorithm 2 in the original submission, we have a typo and missed the $\tilde{W}$, and we have corrected it in the revised manuscript.

---

> ### Author Response · Authors · 2024-11-21
>
> Thanks for your comments. We have incorporated the revised parts into **our revised manuscript**. Please feel free to continue the discussion if anything remains unclear.

---

> > ### Author Response · Authors · 2024-11-25
> >
> > Dear Reviewer,
> >
> > We believe that the additional information we provided in our rebuttal—such as new experimental results, further details, and clarifications on misunderstandings—addresses your key questions. Please let us know if our response has adequately addressed your concerns. We are more than willing to discuss any points that may still be unclear. We hope that the improvements and clarifications provided in our response will positively influence your assessment of our work.
> >
> > Best, Authors of Paper 4022

---

> > > ### Comment · Reviewer_Xe6T · 2024-11-25
> > >
> > > Thanks the authors for the rebuttal. I intend to keep my score unchanged. The result on Autoformer is informative. It would be more interesting to see the effect of adding RBF and ILT to PatchTST and RLinear since those are more competitive method and will give us more intuition about the effect of RBF and ILT.

---

> > > > ### Author Response · Authors · 2024-11-27
> > > > **Further response**
> > > >
> > > > > **Q1.** The effect of adding RBF and ILT to PatchTST and RLinear.
> > > >
> > > > Thanks for your comment. We have conducted experiments of adding RBF and ILT to PatchTST and RLinear. And results are shown in the following Table:
> > > >
> > > > | **Dataset** | **Length** | **PatchTST(MSE)** |   **PatchTST(MAE)**    | **PatchTST(MSE)+RBF** |   **PatchTST(MAE)+RBF**    | **PatchTST+ILT(MSE)** |    **PatchTST+ILT(MAE)**   |
> > > > |:-----------:|:----------:|:------------:|:-----:|:----------------:|:-----:|:----------------:|:-----:|
> > > > | **ETTh1**   | 96         | 0.414        | 0.419 | 0.780            | 0.677 | 0.399            | 0.428 |
> > > > |             | 192        | 0.460        | 0.445 | 0.913            | 0.743 | 0.465            | 0.461 |
> > > > |             | 336        | 0.501        | 0.446 | 0.860            | 0.711 | 0.510            | 0.480 |
> > > > |             | 720        | 0.500        | 0.488 | 0.883            | 0.726 | 0.568            | 0.535 |
> > > > | **ETTh2**   | 96         | 0.302        | 0.348 | 1.338            | 0.874 | 0.359            | 0.394 |
> > > > |             | 192        | 0.388        | 0.400 | 1.383            | 0.883 | 0.486            | 0.526 |
> > > > |             | 336        | 0.426        | 0.433 | 1.415            | 0.892 | 0.538            | 0.499 |
> > > > |             | 720        | 0.431        | 0.446 | 1.401            | 0.890 | 0.912            | 0.673 |
> > > > ||
> > > > | **Dataset** | **Length** | **RLinear(MSE)**  |  **RLinear(MAE)**     | **RLinear+RBF(MSE)**  |  **RLinear+RBF(MAE)**      | **RLinear+ILT(MSE)**  |    **RLinear+ILT(MAE)**   |
> > > > | **ETTh1**   | 96         | 0.386        | 0.395 | 0.501            | 0.469 | 0.384            | 0.402 |
> > > > |             | 192        | 0.437        | 0.424 | 0.537            | 0.490 | 0.429            | 0.426 |
> > > > |             | 336        | 0.479        | 0.446 | 0.567            | 0.507 | 0.462            | 0.445 |
> > > > |             | 720        | 0.481        | 0.470 | 0.565            | 0.528 | 0.463            | 0.463 |
> > > > | **ETTh2**   | 96         | 0.288        | 0.338 | 0.359            | 0.393 | 0.307            | 0.355 |
> > > > |             | 192        | 0.374        | 0.390 | 0.434            | 0.435 | 0.387            | 0.402 |
> > > > |             | 336        | 0.415        | 0.461 | 0.462            | 0.460 | 0.424            | 0.434 |
> > > > |             | 720        | 0.420        | 0.440 | 0.459            | 0.466 | 0.424            | 0.443 |
> > > >
> > > >
> > > > The results show that adding RBF and ILT leads to inconsistent improvements in model performance, as evidenced by varying MSE and MAE values across different lookback lengths. This instability may stem from the redundant attention mechanism and reversible normalization, which do not effectively leverage the advantages of frequency domain analysis. For further details, **please refer to the table included in Section 6.10 of the Appendix in the revised manuscript**.

---

### Author Response · Authors · 2024-11-21
**General response**

We thank the reviewers for their thorough and constructive comments. We are glad that the reviewers recognize that our method is "innovative" (7Gt7, 8Y8C, 6yv6), has "reasonable motivation" (RZwJ), "extensive experiments" (vEmK, RZwJ, 6yv6), "clear writing" (vEmK, 7Gt7), "outstanding empirical performance" (Xe6T),  "enhance Mamba's performance"(vEmK, RZwJ), "solid performance gains" (8Y8C), and "robustness" (6yv6, 7Gt7).

Based on the reviewers' valuable feedback, we have performed additional experiments and updated the manuscript. The major additional experiments and improvements are as follows:

1) We conducted experiments on new Mamba-based baselines, including SST and Bi-Mamba+ in Table 1 on all datasets, according to suggestions of Reviewer vEmK and Reviewer 6yv6. Our method clearly outperforms these two Mamba-based baselines. For more details, please refer to responses to reviewers vEmK and 6yv6.
2) We have calculated Pearson correlation values and shown results in Table 3 in Appendix 6.6, following suggestions of Reviewer RZwJ. Our method outperforms other baselines in most of cases. For more details, please refer to responses to Reviewer RZwJ.
3) We have added more related work on Mamba-based methods for time series prediction in Section 6.3 in Appendix, according to suggestions of Reviewer vEmK. For more details, please refer to responses to Reviewer vEmK.
4) We have edited Figure 1, Figure 6, Figure 7, Figure 13, following suggestions of Reviewer 8Y8C.
5) We have conducted experiments on lookback length 1500 and shown results in Table 4 in Section 6.9 in Appendix, following suggestions of Reviewer RZwJ. Our method outperforms other baselines. For more details, please refer to responses to Reviewer RZwJ.
6) We have conducted experiments on combining RBF and ILT with AutoFormer and shown results in Table 5 in Section 6.10 in Appendix, following suggestions of reviewers Xe6T, vEmK, and vEmK. For more details, please refer to responses to Reviewer Xe6T and vEmK.
7) We have conducted experiments on computational overhead comparison and shown results in Table 6 in Section 6.11 in Appendix, following suggestions of Reviewer vEmK, Reviewer 7Gt7, Reviewer RZwJ and Reviewer 8Y8C. For more details, please refer to responses to Reviewer RZwJ, 7Gt7, and 8Y8C.
8) We have conducted experiments on replacing the RBF kernel with Laplacian kernel and Sigmoid kernel and shown results in Table 7 in Section 6.12 in Appendix, according to suggestions of Reviewer 8Y8C and Reviewer 6yv6. Our kernel outperforms the other two kernels. For more details, please refer to responses to Reviewer 8Y8C and 6yv6.
9) We also provided further explanations on questions of all reviewers.

Your time and effort in reviewing our paper are deeply appreciated. The constructive feedback from all the reviewers has been incorporated in our revised manuscript.

---

### Meta-Review · Area_Chair_yEKu · 2024-12-21

**Metareview:**

The paper introduces FLDmamba, a new Mamba-based Forecasting model. Adapting Mamba-style SSMs to the Time Series Space is certainly an interesting direction that is worth investigating, and the reviewers appreciated the enhancements to Mamba's performance on time series tasks by incorporating RBF, Fourier, and Laplace Transform Decomposition and the extensive experiments and ablation studies using popular benchmark datasets. While this is a well written paper with promising experimental results, several authors expressed some reservations (both during and after the rebuttal process) around two points. First the RBF, Fourier, and Laplace Transform blocks and the challenges they purport to address (multi-scale periodicity, transient dynamics, and data noise) seem very orthogonal to the choice of Mamba as the backbone architecture. This weakens the focus and motivation of the paper. The authors' empirical observations (during the rebuttal) that these enhancements did not significantly help non-Mamba architectures seems counter-intuitive and would benefit from a more thorough investigation. Secondly, several reviewers questioned whether the results presented are really SOTA, given that they do not outperform zero-shot models like MOIRAI. (Another observation that reinforces this question is that the original PatchTST paper reports much stronger results than the PatchTST results in this paper). While zero-shot forecasting is a relatively newer research area, it is still a relevant comparison for a largely empirical paper such as this one.

This was truly a borderline paper, and the decision took into account post-rebuttal discussions with the reviewers. The AC urges the authors to revise the paper based on the above reviewer concerns and resubmit to a future venue.

**Additional Comments On Reviewer Discussion:**

Several reviewers raised questions around whether the RBF, Fourier, and Laplace Transform enhancements were truly specific to Mamba and whether they could be applied to other models. The authors did perform experiments by adding these enhancements to other baselines and showcasing little to no benefits. However the reasoning around why these improved FLDMamba but not the baselines significantly was not intuitive.

Reviewers also asked for adding new Mamba-based baselines, for replacing the RBF kernel with other kernels, and for benchmarking computational overhead, all of which the authors satisfactorily answered.

Several reviewers (during and after the rebuttal) asked about whether the results presented are really SOTA, given that they are outperformed by  zero-shot models like MOIRAI. While zero-shot and full-shot models remain separate research directions, it is still a relevant comparison for an empirical paper such as this one.  Another observation that reinforces this SOTA concern is that the original PatchTST paper reports stronger results than what the paper's PatchTST baseline reported.

---

### Decision · Program_Chairs · 2025-01-22

Reject